EMBO
reports

# Perforation of the host cell plasma membrane during *Toxoplasma* invasion requires rhoptry exocytosis

Frances Male[1,6], Yuto Kegawa [2,6], Paul S Blank [2], Irene Jiménez-Munguía [2], Saima M Sidik[3], Dylan Valleau[3], Sebastian Lourido [3,4], Maryse Lebrun[5], Joshua Zimmerberg [2✉] & Gary E Ward [1✉]

## Abstract

***Toxoplasma gondii* is an obligate intracellular parasite. Proteins released during host cell invasion from apical secretory organelles known as rhoptries are delivered into the host cell cytosol to perform functions critical for parasite survival and virulence. How these effector proteins move across the host cell plasma membrane is unknown but may involve a previously noted temporary loss of host cell plasma membrane barrier integrity. Here, we use high-speed, multi-wavelength fluorescence imaging to spatially monitor the barrier integrity of the host cell plasma membrane, in real time, during invasion. The data reveal that early in invasion the parasite creates a transient perforation in the host cell membrane. The perforation occurs at the point on the host membrane in contact with the parasite's apical end. Parasites depleted of any of five proteins known to be required for rhoptry exocytosis are unable to perforate the host cell membrane. These data suggest a model in which perforating agents stored within rhoptries are released onto the host cell at the initiation of invasion to create a conduit for the delivery of rhoptry effector proteins.**

**Keywords** *Toxoplasma*; Apicomplexa; Parasite; Invasion; Rhoptry
**Subject Categories** Membranes & Trafficking; Microbiology, Virology & Host Pathogen Interaction

See also: Y Kegawa et al

## Introduction

Invasion of host cells is a critical step in the lytic cycle of the protozoan parasite *Toxoplasma gondii* and is essential for establishing infection. The overall process of invasion is well conserved among apicomplexan parasites, including *T. gondii* and the related parasites that cause malaria (*Plasmodium* spp.) and cryptosporidiosis (*Cryptosporidium* spp.). Invasion involves the recognition of and attachment to a host cell, delivery of parasite proteins into that host cell, and parasite-driven internalization. Prior to invasion, the proteins to be translocated into the host cell are stored in the rhoptries, which are club-shaped exocytic organelles that consist of a posterior bulb and an elongated neck docked at the apical end of the parasite. Delivery of rhoptry proteins into the host cell is essential for invasion. For example, translocated rhoptry neck proteins (RONs) form a complex at the host cell plasma membrane that serves as a binding site for the parasite, providing traction at the "moving junction" through which the parasite propels itself into the host cell during invasion (Alexander et al, 2005; Besteiro et al, 2009; Lebrun et al, 2005). Translocated rhoptry bulb proteins (ROPs) act downstream of invasion to suppress the innate immune response and manipulate the host cell in ways that facilitate parasite intracellular survival and replication (Butterworth et al, 2022; Fukumoto et al, 2021; Hernández-de-Los-Ríos et al, 2019; Kochanowsky et al, 2021; Li et al, 2020; Steinfeldt et al, 2010). Given their key biological functions, it is not surprising that rhoptry effector proteins are crucial for parasite virulence (Saeij et al, 2006; Shwab et al, 2016; Taylor et al, 2006). After invasion, intracellular parasites send a "second wave" of effector proteins from a different set of secretory organelles (the dense granules) across the parasitophorous vacuole membrane and into the host cell cytosol. These dense granule proteins also serve multiple functions within the host cell, including regulation of the activity of the translocated rhoptry effector proteins, to produce a favorable intracellular environment for parasite growth (Griffith et al, 2022; Panas and Boothroyd, 2021; Seizova et al, 2024).

The rhoptry exocytic machinery has become increasingly well defined in recent years. A search for apicomplexan orthologs of genes important for the discharge of exocytic organelles in ciliates, fellow members of the Alveolate superphylum, has identified a growing set of proteins important for rhoptry exocytosis (Aquilini et al, 2021; Sparvoli et al, 2022). Through reverse genetics and in situ cryo-electron microscopy, the detailed structures of the rhoptry secretory apparatus (RSA) and many of its molecular components have been determined in *Toxoplasma*, *Plasmodium*, and *Cryptosporidium* (Aquilini et al, 2021; Gui et al, 2023; Mageswaran et al, 2021; Martinez et al, 2022; Segev-Zarko et al, 2022; Sun et al, 2024). The RSA lies at the extreme apical tip of the

[1]Department of Microbiology and Molecular Genetics, University of Vermont Larner College of Medicine, Burlington, VT, USA. [2]Section on Integrative Biophysics; Division of Basic and Translational Biophysics, Eunice Kennedy Shriver National Institute of Child Health and Human Development (NICHD), National Institutes of Health (NIH), Bethesda, MD, USA. [3]Whitehead Institute, Cambridge, MA, USA. [4]Biology Department, Massachusetts Institute of Technology, Cambridge, MA, USA. [5]LPHI, CNRS, INSERM, Université de Montpellier, 34095 Montpellier, France. [6]These authors contributed equally: Frances Male, Yuto Kegawa. ✉E-mail: zimmerbj@mail.nih.gov ; gary.ward@uvm.edu

parasite plasma membrane and forms the apical rosette, a transmembrane structure to which an underlying membrane-bound apical vesicle (AV) is typically docked (Aquilini et al, 2021; Paredes-Santos et al, 2011; Porchet-Hennere and Nicolas, 1983). The AV links the RSA to the tips of the rhoptries (Aquilini et al, 2021; Mageswaran et al, 2021). Exocytosis of lumenal rhoptry proteins through the rosette therefore involves two membrane fusion events: the rhoptry tip with the AV, and the AV with the parasite plasma membrane. The order in which these fusion events take place and whether they are independent or co-regulated is currently unknown, although recent ultrastructural data from *Plasmodium* merozoites suggest that rhoptry-to-AV fusion occurs first (Martinez et al, 2022).

Despite these advances in our understanding of the composition and ultrastructure of the RSA, the mechanism by which the exocytosed rhoptry proteins are translocated into the host cell remains unknown. There is no evidence that the MYR1/2/3-based machinery involved in the translocation of dense granule proteins across the parasitophorous vacuole membrane (Franco et al, 2016; Marino et al, 2018; Panas and Boothroyd, 2021; Seizova et al, 2024) plays any role in the earlier events of rhoptry effector translocation. The uncharacterized mechanism underlying rhoptry effector translocation is likely unique to *T. gondii* and related apicomplex-ans. Other parasites deliver cargo into target cells via exosomes, often at a distance (Montaner et al, 2014), but there is no evidence that such a process occurs in *T. gondii*; rather, the apical localization of the RSA suggests a protein transfer mechanism that requires apposition between the parasite apex and the host cell. Prokaryotic needle-like type III, IV, and VI secretion systems are an alternative mechanism for protein delivery between cells (reviewed in Filloux, 2022; Green and Mecsas, 2016), but *T. gondii* does not possess orthologous gene products, nor have orthologous structures been observed in the many published electron micrographs of parasites prior to or during invasion (Aquilini et al, 2021; Dubremetz, 2007; Dubremetz et al, 1985; Gui et al, 2023; Hakansson et al, 2001; Mageswaran et al, 2021; Martinez et al, 2022; Nichols et al, 1983; Sadak et al, 1988; Segev-Zarko et al, 2022; Sun et al, 2024). Fusion of the parasite and host cell plasma membranes can also be ruled out, since electrophysiology experiments have shown that host cell plasma membrane capacitance – which measures membrane surface area— does not increase during invasion (Suss-Toby et al, 1996) as would be expected if the two membranes were to fuse.

While the electrophysiology experiments referred to above were undertaken to determine host cell membrane capacitance during invasion, a transient increase in host cell membrane conductance was also noted (Suss-Toby et al, 1996), providing a potential clue to the mechanism underlying rhoptry protein translocation. Here, we describe the development of a new high-speed fluorescence microscopy-based assay that allows us to spatially monitor the plasma membrane barrier integrity of many cells, in real time, during invasion by *T. gondii*. This assay confirmed that immediately preceding invasion, the parasite creates a transient perforation in the host cell membrane and demonstrated that the perforation occurs at the point on the host cell membrane in contact with the interacting parasite's apical end. Using this assay, parasites that can be conditionally depleted of five different proteins that function in rhoptry exocytosis were tested for their ability to induce the perforation. In all cases, a block in rhoptry exocytosis led to a block

in host cell perforation. These results are consistent with a model in which material stored within the rhoptries is exocytosed upon contact with the host cell, causing a transient perforation in the host cell membrane through which rhoptry effector proteins are delivered. In a companion paper (Kegawa et al, 2025), the electrophysiological characteristics of the invasion-associated change in host cell plasma membrane conductance are analyzed in detail to explore the nature of the perforation.

## Results

### Host cell invasion by *T. gondii* is associated with a large, transient increase in host cell plasma membrane conductance

COS-1 cells were patch clamped in the whole-cell configuration and held under voltage-clamp conditions ($-60$ mV) to measure current flowing across the host plasma membrane before, during, and after invasion. Invasion was simultaneously visualized using differential interference contrast (DIC) microscopy. A transient increase in host cell membrane conductance was invariably observed early in the invasion process, before moving junction formation becomes apparent as a constriction in the body of the parasite (e.g., Fig. 1), as previously noted (Suss-Toby et al, 1996). Characterization of 25 conductance transients showed them to consist of a rapid increase ($50 \pm 10$ ms) to an average peak amplitude of $3.40 \pm 1.12$ nS, followed by a slower return to near baseline ($197 \pm 39$ ms) (Kegawa et al, 2025).

### Host cell perforation can be visualized as a rapid influx of extracellular calcium into the host cell at the point of apical parasite attachment

Because the patch clamp technique is technically demanding and its throughput is low, we sought to develop an alternative, higher throughput assay to visualize the perforation event. The patch

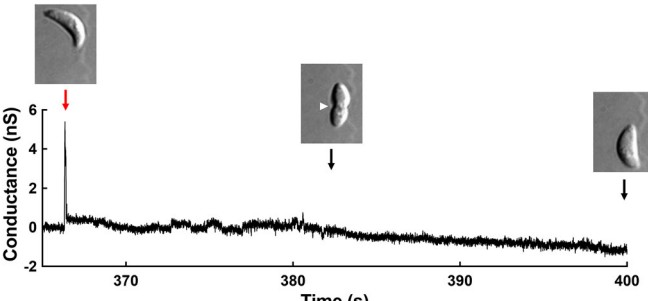

**Figure 1. *T. gondii* invasion is associated with a transient increase in host cell plasma membrane conductance.**

Current across a COS-1 cell membrane was measured under voltage-clamp conditions prior to, during, and after invasion by *T. gondii*. A single large change in host cell conductance (red arrow) is invariably observed immediately before parasite internalization, which is visualized by DIC microscopy as a constriction in the parasite plasma membrane (white arrowhead) as the parasite passes through the moving junction and into the host cell. Source data are available online for this figure.

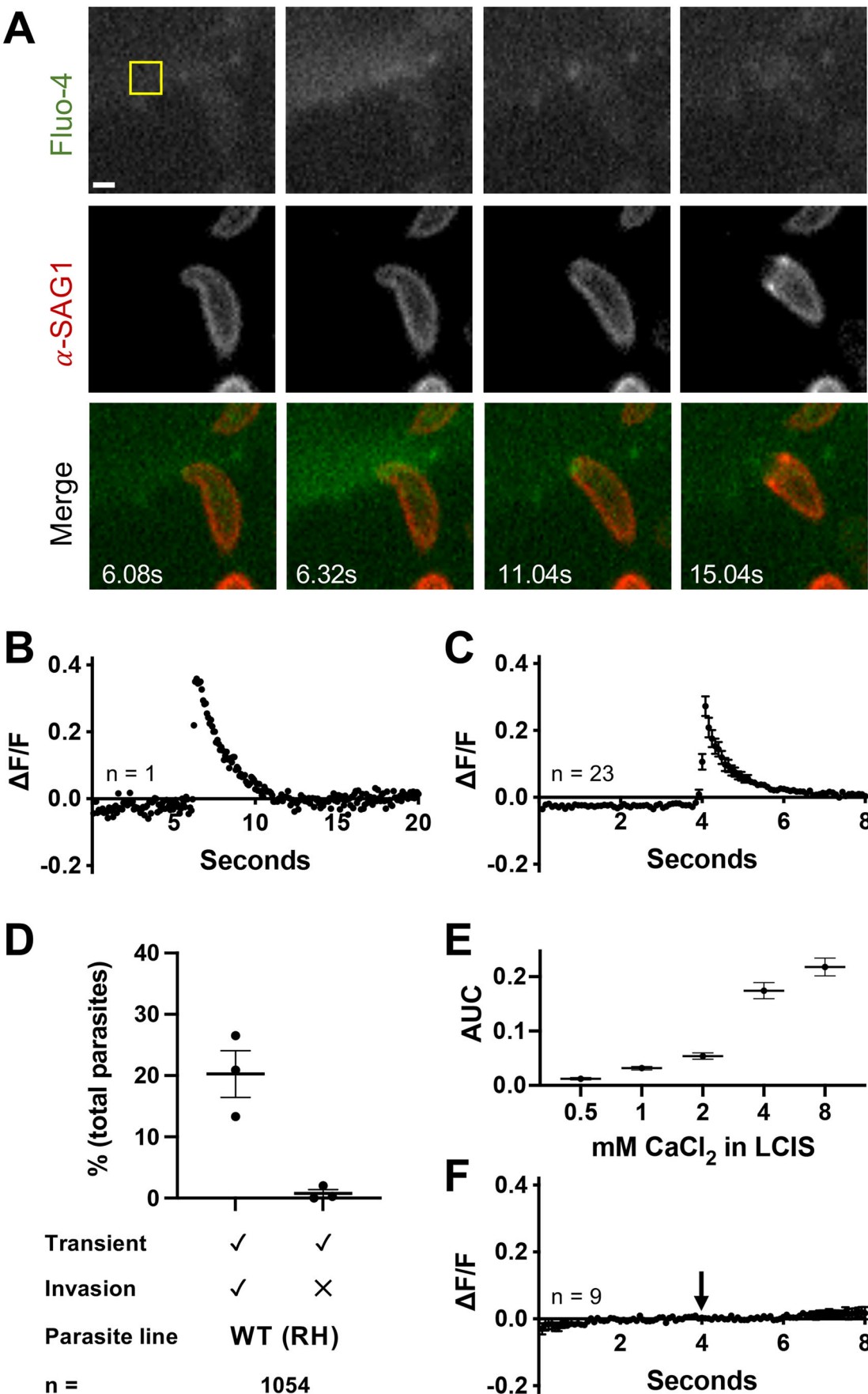

**Figure 2. A transient increase in intracellular calcium is observed within the host cell at the site of *T. gondii* invasion.**

(A) Individual frames from a time series showing changes in Fluo-4 fluorescence and stripping of fluorescently conjugated anti-SAG1 antibody from the surface of invading *T. gondii*. Top panels: Fluo-4 (pseudocolored green in merge); middle panels: anti-SAG1 (pseudocolored red in merge); bottom: merge. Scale bar = 2 μm. The calcium transient reaches maximal intensity at 6.32 s. The full video from which these frames were extracted is presented as Movie EV1. (B) Quantification of Fluo-4 fluorescence intensity in the host cell during the invasion event depicted in (A), within the region of interest marked by the yellow box. (C) Consensus plot of Fluo-4 fluorescence intensities from all invasions captured in a single day (n = 23, four technical replicates). The fluorescence intensities in the 100 frames surrounding the peak of each calcium transient were averaged across all transients, the peaks of which were aligned to frame 51. The plot shows the mean ± SEM at each time point. (D) Quantification of calcium transients induced by WT (RH) parasites and whether they are associated with invasion events. Each data point represents one biological replicate consisting of the average of two to three technical replicates. Total parasite counts are shown in Fig. EV2A; horizontal bars indicate mean ± SEM (n = 1054 parasites). (E) The area under the curve (AUC) was measured for calcium transients generated in different extracellular calcium concentrations. At least 20 transients from two to three biological replicates were measured at each calcium concentration; mean ± SEM. (F) Consensus plot of Fluo-4 fluorescence levels from invasions in calcium-free conditions (n = 9). For each invasion event, 100 frames were plotted and traces were aligned such that invasion began at 4 s (arrow); mean ± SEM. Source data are available online for this figure.

clamp method detects total current through the perforation, a physical measurement. We hypothesized that the perforation should also create a detectable chemical signal, i.e., the influx of a single cation, calcium, down its steep (10,000- to 20,000-fold) concentration gradient through the perforation and into the host cell. Using the intracellular fluorescent calcium indicator, Fluo-4 AM, a transient increase in Fluo-4 fluorescence was indeed observed within the host cell at the site of invasion (Fig. 2A, top row). To unambiguously identify invading parasites, the parasites were pre-labeled with a fluorescently conjugated antibody against the most abundant surface protein, surface antigen 1 (SAG1); as the parasite invades, the bound antibody is stripped from the parasite surface at the moving junction (Dubremetz et al, 1985 and Fig. 2A, middle row). Rapid excitation switching was used for near-simultaneous imaging of the signals from the calcium indicator and the labeled parasites, enabling direct correlation of the calcium transients with specific invading parasites (Fig. 2A, bottom row and Movie EV1). A single calcium transient was observed per invasion event. Furthermore, as suggested by the data in Fig. 2A and demonstrated below at higher spatial resolution, the calcium transients within the host cell initiate at the point of apical parasite attachment, indicating a highly localized perforation event. Quantification of the invasion-associated calcium transient visualized in Fig. 2A is shown in Fig. 2B. While there was some variability between individual calcium transients (Appendix Fig. S1), aligning the transients by their maximal intensity reveals that the overall magnitude and kinetics of the calcium transients were consistent across independent invasion events (e.g., Fig. 2C, n = 23).

The Fluo-4 fluorescence intensity data shown in Fig. 2B,C were manually extracted from the video sequences, one event at a time. To improve throughput, we developed a semi-automated analysis pipeline that identifies calcium transients in an entire field of view (containing, on average, 165 ± 11 parasites) and enables subsequent visual comparison of the transients to the captured images of invading parasites (see Fig. EV1 and Methods for details). Using these methods, we observed that 21.0 ± 4.4% of the wild-type (WT) RH parasites added to a host cell monolayer induced calcium transients during the 96 s of recording time (Fig. 2D). Of those parasites that induced transients, 98.6% subsequently invaded (Fig. EV2). Large numbers of parasites can be analyzed using these methods (n = 1054 in Fig. 2D), enabling robust statistical comparisons between populations of parasites (Fig. EV2).

To confirm that the calcium transients were due to the influx of extracellular calcium rather than signal-mediated release of calcium from host cell intracellular stores, we conducted experiments with

different concentrations of extracellular calcium. The magnitudes of the invasion-associated calcium transients were proportional to the concentration of calcium in the extracellular medium (Fig. 2E), and transients were not detected during invasion in calcium-free medium (Fig. 2F).

To test whether the conductance and calcium transients are readouts of the same perforation event, i.e., whether the ion flux detected via patch clamp recording occurs through the same pathway as the calcium influx detected using Fluo-4, we compared the two datasets directly using denoised and peak-normalized consensus transients for conductance (Fig. 3A, n = 25) and calcium (Fig. 3B, n = 30). Since ion flux scales with the magnitude of the permeability pathway and its duration, we used the area under the curve (AUC) values of the mean normalized conductance and calcium transients, from the start of the transients to their peaks, to compare their distributions (Fig. 3C). There is no significant difference between these two distributions, based on a two-sample Kolmogorov–Smirnov test (p = 0.51; see Methods for details), consistent with the two signals being correlated.

Taken together, these data (a) demonstrate that the intracellular calcium transients reflect entry of extracellular calcium into the host cell and (b) validate the calcium influx assay as a useful alternative approach to patch clamp electrophysiology for monitoring perforation of the host cell plasma membrane during invasion (see Discussion for further comparison of the two assays).

Having established the calcium influx assay as an alternative way to visualize the perforation, we used this assay and a collection of parasites that can be conditionally depleted of key proteins involved in the different steps of rhoptry exocytosis (Fig. 4A) to test whether rhoptry exocytosis is necessary for host cell perforation during invasion.

## Parasites depleted of CLAMP, which triggers rhoptry discharge in response to host cell binding, generate fewer calcium transients than wild-type parasites

CLAMP (claudin-like apicomplexan microneme protein) is a transmembrane protein involved in generating the signals that mediate rhoptry discharge, likely via recognition of as-yet unidentified molecules on the host cell surface (Sidik et al, 2016; Valleau et al, 2023). Parasites in which CLAMP is conditionally depleted by treatment with rapamycin were defective in both invasion and rhoptry protein transfer into the host cell (Sidik et al, 2016; Valleau et al, 2023). Note that the assays currently available to study rhoptry secretion are end-point assays that measure the

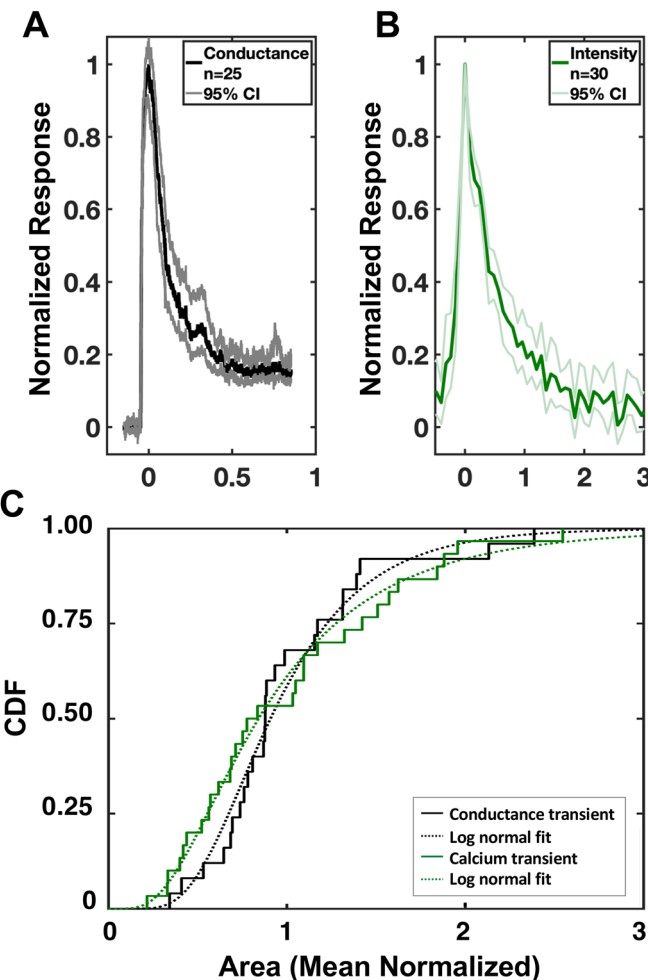

**Figure 3.** **Comparison of the conductance and calcium transients induced in the host cell during invasion.**

(A, B) Consensus transients for (A) conductance (black) with 95% confidence interval (gray; $n = 25$) and (B) Fluo-4 intensity (green) with 95% confidence interval (light green; $n = 30$). The peak of each transient was normalized to 1.0 for comparison. (C) Area under the curve (AUC) data for the conductance (black) and Fluo-4 intensity (green) transients, from the start of the transient to the peak. Mean normalized transients were plotted as their cumulative distribution functions (CDFs), each with a log-normal fit. $p = 0.51$, two-sample Kolmogorov–Smirnov test. Source data are available online for this figure.

presence of rhoptry proteins in the host cell; as such, they do not distinguish between rhoptry exocytosis and transfer of the exocytosed proteins into the host cell. We confirmed rapamycin-dependent CLAMP depletion in these parasites by Western blot (Appendix Fig. S2A). CLAMP-depleted parasites generated fewer calcium transients in the host cell than parasites expressing CLAMP (Fig. 4B). When the data for multiple independent biological replicates were quantified, these differences were found to be significant: CLAMP-expressing parasites induced calcium transients at a rate of $22.5 \pm 2.7\%$, compared to $4.2 \pm 0.5\%$ for CLAMP-depleted parasites (Fig. 4C, $p < 0.0001$; see also Fig. EV2B, B' for total numbers of transients and invasions scored). In contrast to the CLAMP conditional knockdown parasites, rapamycin treatment had no effect on the number of calcium transients

generated by the RH DiCre (Andenmatten et al, 2013; Pieperhoff et al, 2015) parental parasite line (see Methods and Appendix Fig. S3). The $4.2 \pm 0.5\%$ residual invasion and associated transient generation seen in the CLAMP-depleted parasites is consistent with the previously reported large but incomplete block in invasion associated with CLAMP depletion (Sidik et al, 2016). Thus, the loss of CLAMP leads to a reduced ability of the parasite to perforate the host cell.

## Parasites require an intact rhoptry secretory apparatus to perforate the host cell

Next, we focused on three members of the RSA—ferlin 2 (FER2), and two non-discharge proteins (Nd9 and NdP1)— each of which is necessary for membrane fusion between the AV and the parasite plasma membrane (Aquilini et al, 2021; Coleman et al, 2018). Protein depletion following treatment with anhydrotetracycline (ATc) was confirmed by Western blot (Appendix Fig. S2B–D). Parasites depleted of these three proteins all showed a phenotype similar to that of the CLAMP knockdown parasites: a significant reduction in host cell perforation and invasion compared to their respective controls (Figs. 5A–C and EV2C–E, C'–E'; $p < 0.0001$ for each of the three lines, comparing number of calcium transients generated by protein-depleted vs. control groups). The effect of ATc treatment on calcium transient generation by the RH TATi (Meissner et al, 2001) parental parasites was taken into consideration in these analyses (see Methods and Appendix Fig. S3). Thus, parasites depleted of proteins required for fusion of the AV and parasite plasma membrane are deficient in their ability to induce perforation of the host cell plasma membrane.

We made two observations during these experiments that were analyzed further. First, in some of our early experiments, we noted invasions without associated calcium transients (Fig. EV2). This observation seemingly argues against the hypothesis that host cell perforation is required for invasion and is distinctly different from the invariable association of the conductance transients with invasion seen by the highly sensitive patch clamp method (Kegawa et al, 2025; Suss-Toby et al, 1996). To test whether calcium transients were present in these experiments but below the limit of detection of our assay, we developed an alternative assay method with improved sensitivity using a different fluorescent calcium indicator (Cal-520 AM), treatment with probenecid to improve indicator retention, and elevated levels of extracellular calcium to improve signal-to-noise. Retrospective retesting of the CLAMP inducible knockdown parasites using this improved method recapitulated the results attained with the standard method, i.e., CLAMP-deficient parasites generate fewer transients (Fig. EV3A). Importantly, the improved method also detected a reduced number of invasions without accompanying transients: 0.08% of the CLAMP-expressing parasites were scored as invading without transients in the improved assay (95% confidence interval (Wilson score interval) 0.01–0.42; Fig. EV3B) compared to 2.26% using the standard method (95% confidence interval 1.75–2.9%; Fig. EV2B). These results, together with the electrophysiological recordings, suggest that few, if any, parasites invade without first perforating the host cell.

The improved assay also allowed us to spatially map calcium transient initiation and spread in finer detail. As shown in Fig. EV3C–F, the transient is first seen at the very tip of the invading

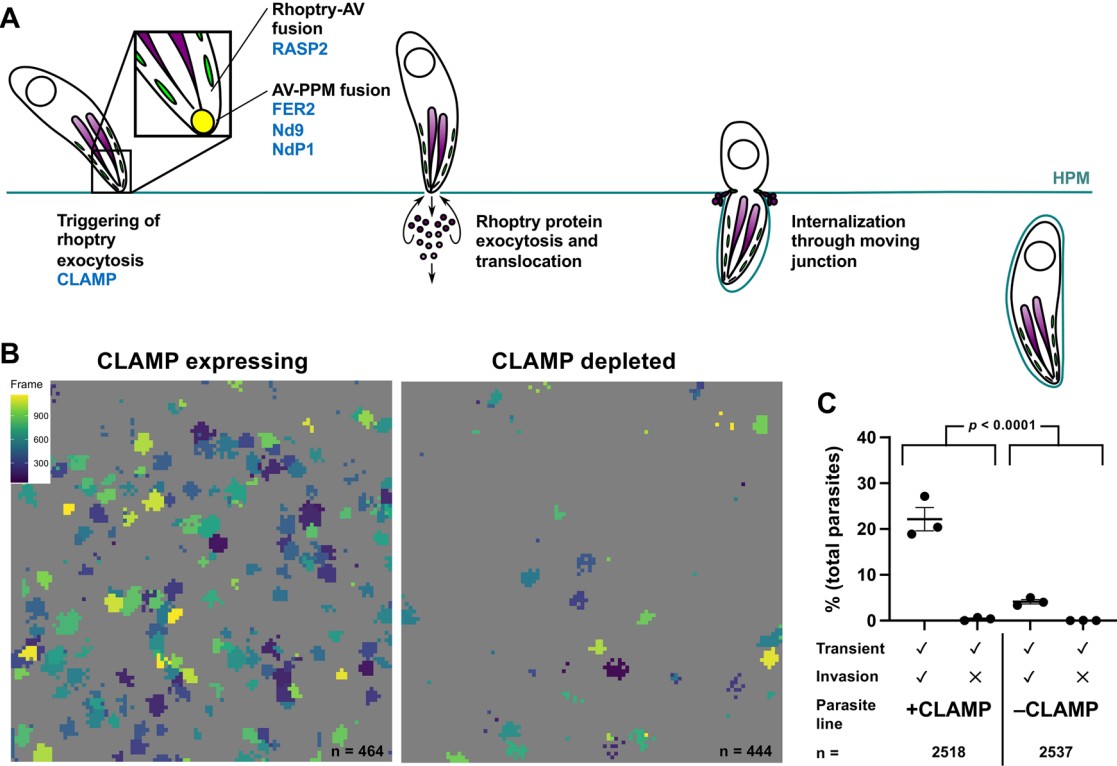

**Figure 4. Parasites depleted of CLAMP generate fewer calcium transients than wild-type parasites.**

(A) Schematic showing the steps of host cell invasion by *T. gondii*, highlighting rhoptry protein exocytosis and protein translocation into the host cell. The rhoptries (purple), micronemes (green), apical vesicle (yellow), and nucleus (open circle) are depicted. Specific proteins known to be involved in rhoptry exocytosis that were evaluated in this study are shown in blue. AV apical vesicle, PPM parasite plasma membrane, HPM host plasma membrane. (B) Representative fields of view (FOV) showing calcium transients generated by CLAMP-expressing (left) and CLAMP-depleted (right) parasites ($n = 464$ and 444 parasites, respectively). Events are color-coded by frame of peak transient intensity (inset, left plot); FOV = $221 \times 221$ μm. (C) Quantification of invasion events and calcium transients induced by control parasites (+CLAMP, $n = 2518$) compared to parasites depleted of CLAMP (−CLAMP, $n = 2537$); horizontal bars indicate mean ± SEM. Each data point represents one biological replicate consisting of the average of three technical replicates. The total number of calcium transients generated by the +CLAMP and −CLAMP groups were compared using Fisher's exact test, $p < 0.0001$ (Fig. EV2B′). Source data are available online for this figure.

parasite and spreads radially outward from that point, dissipating back towards baseline levels within a few hundred milliseconds.

The second observation was that parasites occasionally generate aberrant calcium transients in the host cell. Transients induced by NdP1-depleted parasites are shown in Fig. EV4A as a representative example. A small percentage of the parasites depleted of NdP1 were able to invade, as expected (Aquilini et al, 2021), and these parasites induced transients similar to those generated by WT parasites, as shown in Fig. EV4A, box i. In contrast, two parasites in the same field of view induced aberrant calcium transients (Fig. EV4A, boxes ii and iii), and these parasites failed to subsequently invade. The aberrant transients had a different shape and magnitude and were therefore not well captured by our automatic peak detection software using the parameters originally optimized to maximize peak capture of the transients induced by WT parasites. By altering the peak detection parameters, WT-like transients could still be detected (e.g., Fig. EV4B), and the aberrant transients (e.g., Fig. EV4E,G) were captured at a higher frequency. While the transients associated with invading NdP1-depleted parasites were similar to those generated by WT parasites (compare Fig. EV4D to Fig. 2C), the aberrant transients associated with the non-invading NdP1-depleted parasites were broader, occupying ~10x as many pixels

and showing peak fluorescence intensities approximately twice as large (Fig. EV4E–H; Movie EV2) as the transients associated with invasions (Fig. EV4B,C). These characteristics were consistent across the aberrant transients detected (Fig. EV4I). Using the altered peak detection parameters, the aberrant transients were observed at a low frequency in all parasite lines tested (Fig. EV4J) but were most common with Nd9- and NdP1-depleted parasites ($2.1 \pm 0.6$ and $2.4 \pm 0.5\%$ of the parasites, respectively). The parasites that generated the aberrant transients failed to invade in >93% of the cases monitored (Fig. EV4J), suggesting that the aberrant transients reflect perforations that are defective in some aspect necessary to support invasion.

## Parasites depleted of RASP2, which likely functions in rhoptry-to-AV fusion, are impaired in their ability to generate calcium transients

If the perforation functions in rhoptry protein translocation, the AV would be an ideal compartment in which to sequester the perforating agent, since it could be exocytosed before the bulk of the rhoptry proteins, creating the pathway for translocation across the host cell membrane immediately before the pathway is

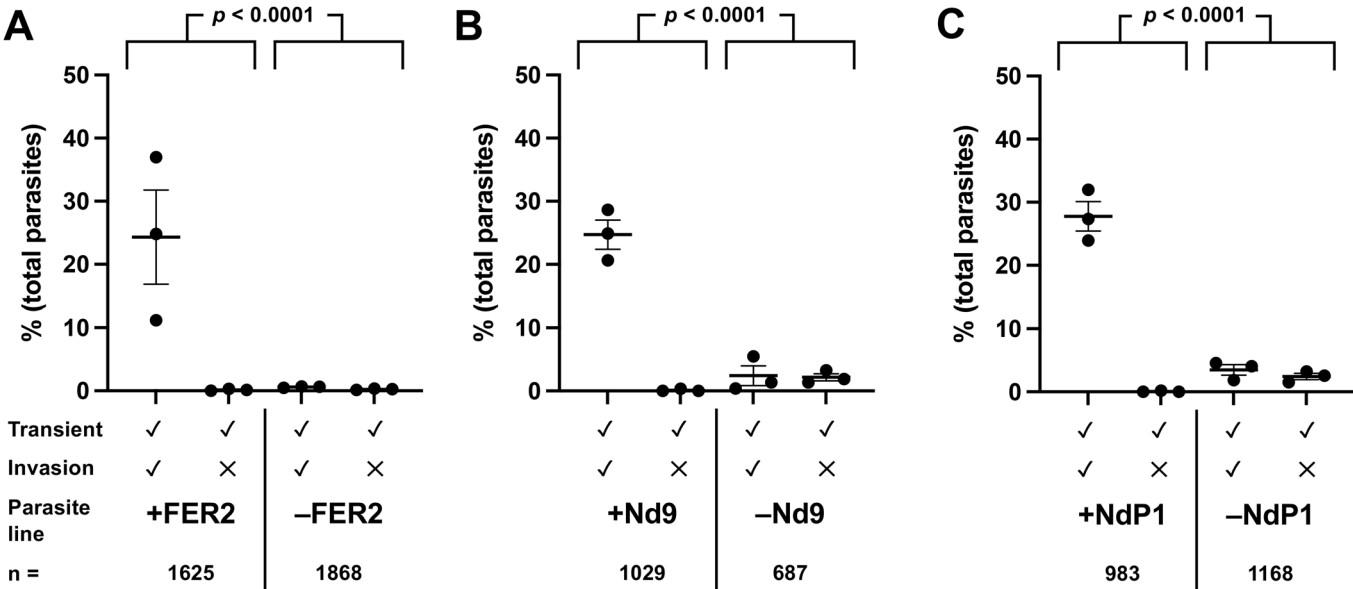

**Figure 5. Parasites with a disrupted rhoptry secretory apparatus generate fewer calcium transients than wild-type parasites.**

(A–C) Quantitative comparison of invasion events and calcium transients induced by mutants depleted of RSA proteins involved in the AV-PPM fusion event (FER2 (**A**), Nd9 (**B**), and NdP1 (**C**)) and their respective controls. Each data point represents one biological replicate, consisting of the average of three technical replicates; horizontal bars indicate mean ± SEM. Total number (*n*) of parasites analyzed per group is indicated. The total number of calcium transients generated by the +protein and -protein groups were compared by Fisher's exact test; *p* < 0.0001 for each of the comparisons shown in (**A–C**), respectively (see also Fig. EV2C'–E'). Source data are available online for this figure.

required. Given that RASP2 (rhoptry apical surface protein 2) is thought to mediate fusion between the rhoptry tip and the AV (Suarez et al, 2019), fusion of the AV to the parasite plasma membrane and release of the perforating agent hypothetically stored within the AV might still occur in RASP2-depleted parasites. RASP2 knockdown in the parasite line used in these experiments was confirmed by Western blot (Appendix Fig. S2E). The RASP2-depleted parasites behaved like those lacking CLAMP or members of the RSA, showing a strong reduction in the number of calcium transients induced (3.1 ± 1.3% compared to 26.4 ± 9.1% for control parasites, Fig. 6A, *p* < 0.0001; see also Fig. EV2F, F' for total numbers of transients and invasions scored). Similarly, RASP2-depleted parasites failed to induce any detectable conductance transients in electrical recordings, while RASP2-expressing parasites induced conductance transients at WT levels (28.8%, 95% confidence interval 20.6–38.2%; Fig. 6B).

## Discussion

We have shown here that the barrier integrity of the host cell membrane is transiently disrupted during the early stages of invasion by *T. gondii*. Given the tight association between the membrane perforation and successful invasion, perforation likely plays an important role in the invasion process. We propose that the perforation provides the pathway through which exocytosed rhoptry effector proteins—including the RON proteins that are critical for moving junction formation and invasion—are delivered into the host cell. Using our new fluorescence microscopy-based assay and a collection of mutant parasites with reduced expression

of proteins involved in the signaling pathway leading to rhoptry exocytosis (CLAMP), fusion of the AV to the parasite plasma membrane (FER2, Nd9, and NdP1), and AV-to-rhoptry fusion (RASP2), we show that rhoptry exocytosis is necessary to generate the perforation. These data suggest that the perforating agent(s) is stored within the rhoptries and/or the AV and is therefore released only upon contact with the host cell, providing a mechanism for the parasite to sequester this potentially harmful material and release it precisely when and where it is needed (Fig. 7).

Host cell perforation was observed by two independent assays: patch clamp electrophysiology and calcium imaging. Several lines of evidence argue that the conductance and calcium transients are experimental readouts of the same perforation event. First, we see exactly one conductance transient and one calcium transient per invasion event. This rules out the possibility that the conductance transient occurs first and triggers a subsequent influx of calcium, because the later calcium influx would have been detectable as a second conductance transient (calcium ions carrying the charge in this case) and a second conductance transient during invasion was never observed. Second, given the enormous calcium electrochemical gradient across the host cell plasma membrane, we would *expect* calcium influx through any but the most highly selective perforations/pores. We note that even the Sec61 channel in the ER membrane, one of the most well-studied and tightly regulated protein translocons, allows some calcium leakage into the cytosol down the steep ER-to-cytosol calcium gradient (e.g., Lang et al, 2017). Conversely, we would expect no calcium transients in a medium lacking calcium, which is also what we observed. Third, while the decay phase of the calcium signal was longer than that of the conductance signal, this too is to be expected if the two assays

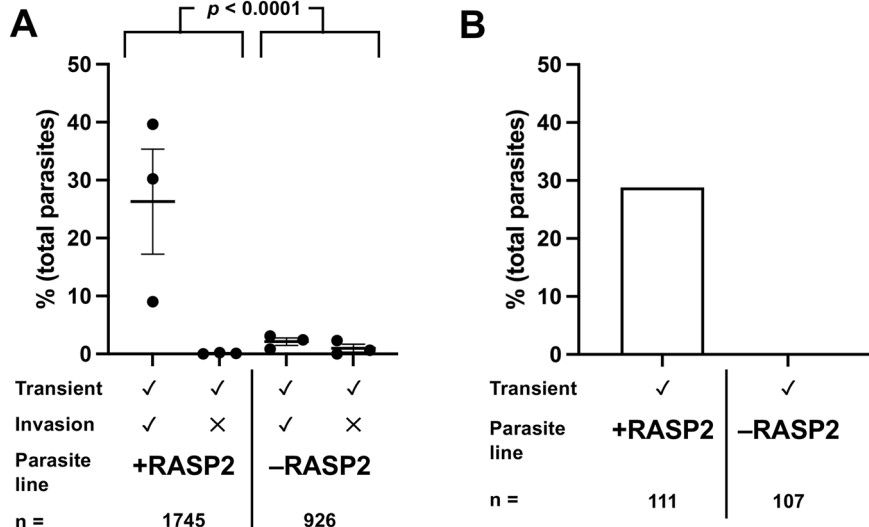

**Figure 6. Parasites depleted of RASP2 generate fewer calcium transients than wild-type parasites.**

(A) Quantification of invasion events and calcium transients induced by control parasites (+RASP2, n = 1745) compared to parasites depleted of RASP2 (-RASP2, n = 926). Each data point represents one biological replicate, consisting of the average of three technical replicates; horizontal bars indicate mean ± SEM. The total number of calcium transients generated by the +RASP2 and −RASP2 groups were compared by Fisher's exact test, p < 0.0001 (Fig. EV2F'). (B) Quantification of the number of conductance transients induced by control parasites (+RASP2, n = 111) compared to parasites depleted of RASP2 (−RASP2, n = 107), as detected by patch clamp. Source data are available online for this figure.

are reporting on the same event. The conductance transients are a direct measure of ion flow through the membrane perforation(s); when the perforation is present, ions flow, and when the perforation closes or is occluded, ion flow stops. The fluorescence-based assay is specifically measuring the number of calcium ions that flow through these same perforations. However, in contrast to patch clamping, which is directly measuring ion flow, the calcium indicator provides a *chemical* signal which, given the on and off rates of the indicator, diffusion of calcium and calcium-bound indicator from the site of calcium entry, and the calcium sequestration mechanisms of the cell, would be expected to result in a longer duration signal than electrophysiological recordings of conductance, which is what we observe. Finally, the distributions of the normalized conductance and calcium signal magnitudes, from initiation of the transients to their peak, are indistinguishable (Fig. 3C), demonstrating that the signals are indeed correlated. The most parsimonious explanation for all these data is that the two assays are reporting on the same event.

The electrical and optical assays are complementary, each with its own unique strengths. Patch clamping provides the most direct way to visualize the perforation, is highly sensitive, and offers excellent (sub-millisecond) time resolution that proved critical to our ability to resolve the transient into discrete unitary conductance events (Kegawa et al, 2025). The calcium indicator assay is less arduous, provides spatial information on the location of the perforation within individual cells, and is higher throughput. The higher throughput of the fluorescence-based assay was critical to our ability to assay multiple mutant parasite lines in sufficiently large numbers (500–2500 parasites per treatment group) for robust statistical comparisons between parasite populations. The sensitivity of the calcium assay can be further improved for future studies using other calcium indicators (e.g., Cal-520 AM), probenecid to

reduce indicator efflux, and (where appropriate) increased extracellular calcium (see Fig. EV3). While transient perforation of the host cell membrane appears to be important for invasion, and calcium influx can serve as a useful experimental readout for monitoring the perforating event, calcium is not itself a major charge-carrying ion in the electrophysiology experiments (Kegawa et al, 2025), nor is calcium influx necessary for invasion (Fig. 2F).

Great strides in our understanding of rhoptry exocytosis have been made in recent years through a comparison of the proteins that regulate rhoptry exocytosis in apicomplexan parasites to those that regulate trichocyst/mucocyst discharge in ciliates (reviewed in Sparvoli and Lebrun, 2021). Ciliates express many homologs of apicomplexan RSA proteins, including FER2, Nd9 and NdP1, and in both lineages, depletion of these proteins leads to defective exocytosis. Similarities and differences in the ultrastructure of the secretory apparatus in the two lineages have also been informative (Aquilini et al, 2021; Mageswaran et al, 2021). One notable difference is the presence in apicomplexans of the AV between the tip of the rhoptry and the parasite plasma membrane: in the two ciliate models (*Paramecium* and *Tetrahymena*), the secretory organelles are docked directly to the plasma membrane. Rhoptry exocytosis in apicomplexans therefore requires two fusion events: rhoptry-to-AV and AV-to-parasite plasma membrane. How might apicomplexans have evolved this extra elaboration, when the exocytotic machinery is otherwise so well conserved? Because the material exocytosed by apicomplexans (but not ciliates) is translocated into another cell, an attractive hypothesis would be that the perforating agent that functions in protein translocation is stored in the AV, as this would provide a mechanism for the perforating agent to be delivered to the host cell membrane before the effector proteins, creating the pathway for their subsequent

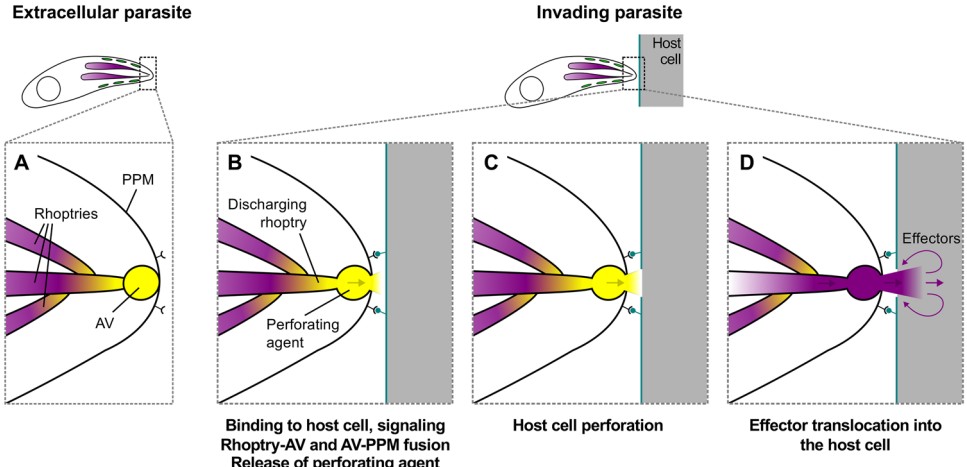

**Figure 7. Model of rhoptry exocytosis-dependent host cell perforation.**

(A) Prior to interaction with the host cell, the perforating agent (yellow) is stored in the AV, the tip of the rhoptries, or both. (B) Binding of parasite surface ligands to receptor(s) on the host cell triggers signaling that leads to rhoptry exocytosis, releasing the perforating agent into the narrow space between the apical tip of the parasite and the host cell. The order in which rhoptry-to-AV and AV-to-PPM fusion take place in *T. gondii* and whether the two fusion events are independent or co-regulated are unknown. (C) A highly localized and transient perforation is created in the host cell membrane. (D) Exocytosed rhoptry effector proteins (purple) emerge from the tip of the parasite immediately after the perforating agent and pass through the perforation and into the host cell.

translocation. In this context, the RASP2 mutant was of particular interest to test in the perforation assay, since RASP2 is thought to mediate rhoptry-to-AV fusion (Suarez et al, 2019) and AV-to-plasma membrane fusion may still occur. However, as with the RSA and signaling mutants, depleting RASP2 disrupted the parasite's ability to perforate the host cell. The simplest explanation for this observation is that the perforating agent is stored within the rhoptries rather than the AV. Alternatively, the two fusion events may not be independent, and blocking one may interfere with the other. Because a perforating agent could potentially damage the AV or rhoptry membrane, it is also possible that the active perforating agent is composed of two inactive components, one stored in the AV and one stored in the neck of the rhoptries. In this scenario, the perforating activity would only be generated when the two compartments mix as a result of rhoptry-to-AV fusion, which in *Plasmodium* likely occurs before exocytosis of the rhoptry contents (Martinez et al, 2022). Resolution of these questions will ultimately require the identification of the perforating agent and its localization within the parasite prior to interaction with a host cell.

Most coccidian parasites, including *Toxoplasma*, possess a characteristic tubulin-based structure at their apical end named the conoid. This cone-shaped cytoskeletal organelle can be repeatedly extended and retracted (Dos Santos Pacheco et al, 2020; Graindorge et al, 2016; Mondragon and Frixione, 1996). Might conoid protrusion by a parasite that is apically attached to the host cell exert a mechanical pushing force on the host cell membrane that causes or contributes to host cell perforation? A least two of the rhoptry exocytosis mutants shown here to have a reduced ability to perforate the host cell (Nd9- and FER2-deficient parasites) are known to support normal calcium ionophore-induced conoid extension (Aquilini et al, 2021; Coleman et al, 2018), so the perforation defect in these mutants (Fig. 5A,B) is unlikely to be due to a secondary effect on conoid extension.

However, we cannot currently rule out a model in which changes in host cell membrane tension induced by conoid extension play some role in either the insertion of the exocytosed perforating agents or the regulation of their function.

The proteins and mechanisms underlying host cell invasion appear to be generally well conserved among apicomplexan parasites. Conserved proteins include those that regulate rhoptry exocytosis, such as RASP2 and Nd9, and depletion of these proteins leads to similar phenotypes in *T. gondii* and *Plasmodium* spp. (Aquilini et al, 2021; Suarez et al, 2019). Intriguingly, a previous report demonstrated that the invasion of Fluo-4-loaded erythrocytes by *P. falciparum* merozoites is occasionally accompanied by "a strong $Ca^{2+}$ flux spreading into the erythrocyte from the invasion site" (Weiss et al, 2015), suggesting that the membrane perforation event that we have described here also occurs during invasion by malaria parasites. In turn, we often observed a phenomenon described in this previous report: after the initial calcium transient, a bright dot of Fluo-4 fluorescence appears at the parasite apex and is visible for a few seconds as the *Toxoplasma* parasite begins to penetrate into the host cell (Fig. EV5; Movies EV1, 3–5). Kymograph analysis confirms that the dot of fluorescence develops after the initial calcium transient (Fig. EV5A) and tracks with the apical tip of the parasite as it enters the host cell (Fig. EV5B). These observations suggest that concomitant with or after the initial perforation of the host cell and influx of calcium, Fluo-4 diffuses in the opposite direction through the perforation and into an interior compartment of the parasite, likely the rhoptries. The diffusing Fluo-4 may have calcium bound to it or may encounter high calcium levels within the parasite, in either case generating a brightly fluorescent dot at the parasite apex.

We also observed a rare but intriguing phenotype where a small subset of the parasites interacting with host cells induced aberrant calcium transients that had larger amplitudes and longer durations

than those typically induced by invading parasites. These aberrant calcium transients were observed most frequently in mutants depleted of proteins involved in fusion of the AV and parasite plasma membrane (Nd9, NdP1) and more rarely in all the other parasite lines studied. Most of these aberrant transients (>93%) were associated with parasites that failed to subsequently invade. Why might the magnitude and kinetics of this subset of transients be different? Closure of the putative pores may require exocytosed rhoptry protein(s) and would therefore be delayed if exocytosis is disrupted or if the proteins are delivered in a less focal, concentrated bolus. Alternatively, if rhoptry effector proteins are normally translocated into the host cell through the perforation, these proteins will likely cause partial occlusion of the putative pore while they are being translocated, thereby reducing calcium influx. In the case of the aberrant transients, perhaps perforation occurs without the normal levels of effector exocytosis, reducing occlusion of the putative pores and resulting in more detectable calcium entry over a longer period.

In summary, the data presented here show that rhoptry exocytosis is required for parasite-induced perforation of the host cell membrane during invasion, providing a mechanism for the parasite to release the perforating agent precisely when and where it is needed. The accompanying paper (Kegawa et al, 2025) begins to address the nature of the perforating agent through a detailed electrophysiological characterization of the parasite-induced perforation event. Efforts are currently underway to identify the perforating agent itself, which will enable a direct test of the hypothesis that the perforation serves as the conduit through which rhoptry effector proteins are delivered into the host cell. Given the central role played by many of the secreted rhoptry effector proteins in parasite virulence (Saeij et al, 2006; Shwab et al, 2016; Taylor et al, 2006), elucidating the mechanism(s) underlying effector entry is not only of fundamental cell biological interest, but may also identify new targets and inspire new strategies for therapeutic development. Rather than targeting a single translocated rhoptry protein, targeting the rhoptry protein delivery mechanism will simultaneously disrupt transfer into the host cell of many of the effector proteins that play a central role in the pathogenesis of the devastating diseases caused by *T. gondii* and other apicomplexan parasites.

## Methods

### Reagents and tools table

| Reagent/resource | Reference or source | Identifier or catalog number |
|---|---|---|
| **Experimental models** | | |
| CCD-1112Sk; foreskin fibroblast (*Homo sapiens*) | ATCC; authenticated by STR profiling | CRL-2429 |
| COS-1; kidney fibroblast (*Cercopithecus aethiops*) | ATCC | CRL-1650 |
| *T. gondii*: strain RH | Gift from Dr. Alan Sher, NIH | |

| Reagent/resource | Reference or source | Identifier or catalog number |
|---|---|---|
| *T. gondii*: strain DiCre/ CLAMP cKD | Sidik et al, 2016 | |
| *T. gondii*: strain TATi/ FER2 cKD | Coleman et al, 2018 | |
| *T. gondii*: strain TATi/ Nd9 cKD | Aquilini et al, 2021 | |
| *T. gondii*: strain TATi/ NdP1 cKD | Aquilini et al, 2021 | |
| *T. gondii*: strain TATi/ RASP2 cKD | Suarez et al, 2019 | |
| **Antibodies** | | |
| Mouse monoclonal anti-SAG1 (clone DG52) | Gift from Dr. David Sibley Origin: Burg et al, 1988 | |
| Rabbit polyclonal anti-ACT1 | Gift from Dr. David Sibley Origin: Dobrowolski et al, 1997 | |
| Mouse monoclonal anti-HA.11 | BioLegend | 901514 |
| Mouse monoclonal anti-Myc (clone 9E10) | Thermo Fisher Scientific | 13-2500 |
| Alexa Fluor 647 Antibody Labeling Kit | Thermo Fisher Scientific | A20186 |
| **Chemicals, enzymes and other reagents** | | |
| Molecular Probes, Fluo-4, AM, cell permeant | Thermo Fisher Scientific | F14201 |
| Molecular Probes, Powerload concentrate | Thermo Fisher Scientific | P10020 |
| AAT Bioquest Cal-520, AM | Thermo Fisher Scientific | NC0831165 |
| Molecular Probes Probenecid, water-soluble | Thermo Fisher Scientific | P36400 |
| Sigma Aldrich Calcium ionophore A23187 | Thermo Fisher Scientific | 50-176-5967 |
| Anhydrotetracycline | Takara Bio | 631310 |
| Invitrogen Rapamycin | Thermo Fisher Scientific | PHZ1235 |
| **Software** | | |
| NIS Elements v. 5.11 | Nikon | |
| FIJI | https://imagej.net/ | |
| MATLAB | MathWorks https://www.mathworks.com | |
| PeakCaller | Artimovich et al, 2017 https://hussmanautism.org/resources/software/ | |
| R, version 4.0.1 | R Foundation for Statistical Computing http://www.R-project.org | |
| Prism, version 10.4.1 | GraphPad http://www.graphpad.com | |
| **Other** | | |
| ibidi μ-Slide VI 0.4 | Thermo Fisher Scientific | 50-305-784 |

## Parasite and cell culture

T. gondii parasite lines were propagated by serial passage in confluent monolayers of human foreskin fibroblast (HFF) cells. HFFs were maintained in Dulbecco's Modified Eagle Medium (DMEM) (Life Technologies, Carlsbad, CA) with 10% v/v heat-inactivated fetal bovine serum (FBS) (Life Technologies, Carlsbad, CA), 10 mM HEPES pH 7.0, 100 units/mL penicillin, and 100 µg/mL streptomycin. Prior to parasite passage, the medium was replaced with DMEM with 1% v/v FBS, 10 mM HEPES pH 7.0, 100 units/mL penicillin, and 100 µg/mL streptomycin.

DiCre/CLAMP parasites were treated with 50 nM rapamycin (Invitrogen, Waltham, MA) to induce CLAMP depletion or an equivalent volume of DMSO for 2 h prior to 48-h culture in drug-free medium (Sidik et al, 2016). All other inducible knockdown (iKD) parasite lines were pretreated with 1.5 µg/mL anhydrotetracycline (ATc) (Takara Bio, San Jose, CA) to induce gene knockdown or an equivalent volume of 100% ethanol (EtOH) for the following time courses prior to experiments: FER2 iKD, 96 h (Coleman et al, 2018); Nd9 iKD, 72 h (Aquilini et al, 2021); NdP1 iKD, 72 h (Aquilini et al, 2021); and RASP2 iKD, 48 h (Suarez et al, 2019). Experiments with Nd9 iKD parasites were performed in the continuous presence of 1.5 µg/mL ATc or EtOH.

To isolate parasites for experiments, DMEM containing 1% FBS was replaced with Endo buffer (Endo et al, 1987) modified to include calcium (44.7 mM $K_2SO_4$, 8 mM $MgSO_4$, 2 mM $CaSO_4$, 106 mM sucrose, 5 mM glucose, 20 mM Tris-$H_2SO_4$, 3.5 mg/mL BSA), before HFFs containing large vacuoles were detached from the flask using a cell scraper and parasites released by two to three passages through a 26 G 1/2" blunt needle. Parasites were isolated from cell debris by passage through a Whatman Nuclepore Track-Etch Membrane 3 µm filter (MilliporeSigma, Burlington, MA), and spun at $1000 \times g$ for 2 min. Parasites were then resuspended in 1:20 anti-SAG1 antibody (monoclonal antibody DG52, a generous gift from Dr. David Sibley, 0.2 mg/mL stock; Burg et al, 1988) conjugated to Alexa Fluor 647 (Alexa Fluor™ 647 Antibody Labeling Kit (Molecular Probes, Eugene, OR)) in Endo buffer for 30 min at ambient temperature, then spun at $1000 \times g$ for 2 min, and resuspended in Endo buffer at $3 \times 10^7$ parasites/mL.

## Western blot analysis

Western blots were prepared as described previously (Kelsen et al, 2023). Blots were probed with mouse monoclonal anti-HA.11 antibody 16B12 (BioLegend, San Diego, CA) or mouse anti-Myc monoclonal antibody 9E10 (Developmental Studies Hybridoma Bank, Iowa City IA) diluted 1:1000 along with rabbit polyclonal anti-TgActin antibody generously provided by Dr. David Sibley (Dobrowolski et al, 1997) diluted 1:10,000 as a loading control. LICOR goat anti-mouse and anti-rabbit secondary antibodies (LICOR Biosciences, Lincoln, NE) were diluted 1:20,000.

## Live calcium transient/invasion assay

### Assay preparation

HFFs were seeded in three chambers of an ibidi µ-Slide VI 0.4 (ibidi GmbH, Gräfelfing, Germany) to an average final density of 90% in DMEM containing 10% FBS and incubated for 2 h at 37 °C (with 5% $CO_2$ and humidity). HFFs were washed 1× with live cell imaging solution (LCIS) (Ringer's solution with glucose to 20 mM: 155 mM NaCl, 3 mM KCl, 2 mM $CaCl_2$, 1 mM $MgCl_2$, 3 mM $NaH_2PO_4$, 10 mM HEPES, 20 mM glucose) before being loaded with 5 µM Fluo-4 AM (Invitrogen, Waltham, MA) with 1% PowerLoad Concentrate, 100X (Invitrogen, Waltham, MA) in LCIS for 50 min at ambient temperature. Cells were washed 1× with LCIS to wash out excess indicator and incubated for an additional 40 min at ambient temperature to allow complete de-esterification of intracellular AM esters.

For Fig. EV3, Movie EV4, and Movie EV5, the assays were performed as above except that HFFs were loaded with 5 µM Cal-520 AM (AAT Bioquest, Pleasanton, CA) with 1% PowerLoad Concentrate, 100X (Invitrogen, Waltham, MA), and 1 mM probenecid (Invitrogen, Waltham, MA) for 90 min at 37 °C followed by 30 min at ambient temperature. All subsequent solutions included 1 mM probenecid. Cells were washed 1× with LCIS to wash out excess indicator.

### Imaging

Experiments were carried out at 35–36 °C. Fluo-4-loaded cells were washed 1× with Endo buffer before allowing pre-labeled parasites to settle on cells for 10 min. The buffer was then exchanged for pre-warmed invasion-permissive LCIS to capture calcium transients and invasion events.

For Fig. 2F, the assays were performed as above except for the following changes to yield calcium-free conditions. After antibody labeling, parasites were resuspended in original Endo buffer (Endo et al, 1987) which does not include calcium. After allowing the parasites to settle on cells, this buffer was exchanged for LCIS lacking $CaCl_2$ and containing 1 mM EGTA.

Imaging was carried out on a Nikon Eclipse TE300 widefield epifluorescence microscope (Nikon Instruments, Melville, NY) using a 60× PlanApo λ objective (0.22 µm/pixel, NA 1.4). 1020 × 1020 pixel images were captured using an iXon 885 EMCCD camera (Andor Technology, Belfast, Ireland) set to trigger mode, with exposure time of 39 ms, no binning, readout speed of 30 MHz, conversion gain of 3.8×, and EM gain of 300. Perforation of host cells by parasites resulting in calcium transients and subsequent invasion events were observed by near-simultaneous excitation of Fluo-4 or Cal-520 (490 nm) and Alexa Fluor 647 (635 nm) using a pE-4000 LED illumination system (CoolLED, Andover, England), through rapid excitation switching triggered by the NIS Elements Illumination Sequence module (Nikon Instruments, Melville, NY). Hardware was driven by NIS Elements v. 5.11 software (Nikon Instruments, Melville, NY).

### Data processing

From each capture, 1200 high-quality frames (96 s) were processed. Mean fluorescence intensity (MFI) was extracted from 10,404 10 × 10 pixel regions of interest (ROIs) over time from each 1020 × 1020 pixel field of view (FOV) in ImageJ. In Fig. EV1A–C, a 100 × 100 pixel FOV containing a single calcium transient is shown for clarity. MFI results were passed through the PeakCaller script in MATLAB (MathWorks, Natick, MA) for automated identification of intracellular calcium transients (Artimovich et al, 2017). Using RStudio, calcium transient results from each ROI were plotted back to ROI location and color-coded, based on both the time (frame) the transient reached its peak (Fig. EV1B), and the amplitude (peak) of the transient (Fig. EV1C). Calcium transient and invasion

events were quantified through comparison of identified peaks and live image captures (Fig. EV1D, 600 of 1200 frames shown for clarity). For mutant parasite experiments, three biological replicates were carried out for each mutant. Each biological replicate consisted of three technical replicates for protein-depleted mutants and three technical replicates for controls, carried out on the same day. Samples were not blinded prior to analysis. For the WT RH parasite experiments (Fig. 2D), one technical replicate from one biological replicate was excluded from downstream analysis due to poor quality.

To improve the spatial mapping of the transients, the data were processed as above to show the amplitude (peak) of each transient except that each $100 \times 100$ pixel FOV was divided into $4 \times 4$ pixel ROIs (Fig. EV3C,D). To spatially and temporally show the growth and decay of these transients, the change in fluorescence intensity within each ROI was divided by the median intensity within the ROI over the entire time series ($\Delta F/F$, within the set threshold of 0.2 to 0.8 $\Delta F/F$), and plotted for each ROI within the FOV for the consecutive time points indicated, starting at the frame before the transient could be detected (Fig. EV3E,F).

To better visualize the appearance and movement of the dot of fluorescence at the parasite apex, representative kymographs are shown along paths that bisect the dot and along nearby control paths (Fig. EV5B,D). Kymographs were built using the KymographBuilder plugin in ImageJ.

### Statistical analysis

For each mutant parasite line, differences between protein-depleted parasites compared to controls with respect to induction of calcium transients and invasions were assessed using Fisher's exact test for contingency table analysis. The null hypothesis in each case was that control and protein-depleted parasites would show no differences in the counts observed for each category. The data from three biological replicates were combined into four categories (+transient/+invasion, +transient/−invasion, −transient/+invasion, and −transient/−invasion) and compared using $4 \times 2$ contingency tables (Fig. EV2B–F).

The specific difference in the frequency of calcium transients induced by protein-depleted parasites compared to controls was assessed using Fisher's exact test for the analysis of $2 \times 2$ contingency tables. The null hypothesis in each case was that control and protein-depleted parasites are equally likely to induce calcium transients. The data from the three biological replicates were combined, and the number of transients generated (by invading and non-invading parasites combined) were compared using $2 \times 2$ contingency tables (Fig. EV2B'–F'). Differences reported in the text as "statistically significant" refer to comparisons in which the null hypothesis was rejected based on the reported $p$-value.

Fisher's exact test was also applied to $4 \times 2$ and $2 \times 2$ contingency tables in which counts were corrected based on the percent confluency of the host cell monolayer for each technical replicate. Zero values were treated conservatively (Jovanovic and Levy, 1997) (3/n adjustment). The resulting $p$ values remained consistent with the uncorrected count data (Fig. EV2). Additionally, Fisher's exact test was carried out on $4 \times 2$ and $2 \times 2$ contingency tables, both with and without host cell confluency correction, for each biological replicate individually. For all tests,

$p < 0.0001$, except for one Nd9 biological replicate $2 \times 2$ host cell confluency-adjusted contingency table for which $p = 0.0051$.

## Control experiments were conducted on parental cell lines

Control live calcium transient/invasion experiments were conducted on the RH DiCre parental line after rapamycin or DMSO treatment as described above. Parasites treated with rapamycin or DMSO both induced calcium transients and invaded (Appendix Fig. S3A,B). Rapamycin treatment did not have a significant effect on the induction of calcium transients ($p = 0.75$, Fisher's exact test; Odds ratio (95% CI): 1.02 (0.85, 1.22); Appendix Fig. S3B').

Control live calcium transient/invasion experiments were conducted on the RH TATi parental line after ATc or ethanol (EtOH) treatment as described above. Parasites treated with ATc or EtOH both induced calcium transients and invaded (Appendix Fig. S3C,D). ATc treatment was found to have a significant effect on the induction of calcium transients ($p < 0.0001$, Fisher's exact test; Odds ratio (95% CI): 0.57 (0.44, 0.73), $p < 0.0001$; Appendix Fig. S3D'). To determine the effect of this result on the TATi inducible knockdown (iKD) lines, the Breslow-Day test was used to analyze the interaction between the odds ratio from the TATi control experiment to the odds ratio from each of the iKD line experiments, with the null hypothesis that the odds ratios are equal. For all comparisons, the odds ratios were found to be significantly different ($p < 0.0001$, Breslow-Day test; Appendix Fig. S3E). Further, by converting these odds ratios to effect sizes, the effect size for ATc treatment in the TATi dataset was found to be small-to-moderate (0.2–0.5; Appendix Fig. S3E) while the effect sizes for gene knockdown within all of the iKD datasets were all found to be large (>0.5; Appendix Fig. S3E)(Chinn, 2000). Due to these large differences in effect size, we conclude that the differences observed within each iKD line dataset and presented in the figures are largely due to gene knockdown.

## Electrophysiology experiments

COS1 cells (ATCC CRL-1650) were cultured in DMEM supplemented with high glucose, 200 mM Glutamax, 1 mM sodium pyruvate (Thermo Fisher Scientific, Cat#10569, Waltham, MA) and 100 µg/ml primocin (InvivoGen Cat#ant-pm-2, San Diego, CA) with 10% FBS Premium Select (R&D Systems, Cat#S11550, Minneapolis, MN) at 37 °C under 5% $CO_2$. For experiments, COS-1 cells were seeded onto a 35 mm DT dish (Bioptechs, Butler, PA) at a concentration of $4 \times 10^4$ cells/mL in 1 mL DMEM for 60 min at 37 °C under 5% $CO_2$. before the medium was replaced by LCIS. Temperature was maintained at 37 °C using a Delta T4 Culture Dish Controller (Bioptechs, Butler, PA). The pipette solution for whole-cell recordings contained 122 mM KCl, 2 mM $MgCl_2$, 11 mM EGTA, 1 mM $CaCl_2$, 5 mM HEPES (pH 7.26, adjusted with KOH). Patch pipettes (3 MΩ resistance) were fabricated from 1.5-mm thick wall borosilicate glass capillaries using a P1000 puller (Sutter Instruments, Novato, CA). Whole-cell formation was monitored with an AxoPatch200B amplifier (Molecular Devices, San Jose, CA) in the voltage-clamp mode. The output current was filtered using the internal 100 kHz Lowpass Bessel filter included in the amplifier and an external 5 kHz low

pass, 8 pole, Bessel filter (Model 900 CT/9 L8L, Frequency Devices Inc, Haverhill, MA) and was digitized at 100 μs for a time resolution of 200 μs. A −60-mV holding potential was applied to monitor current changes during interactions with *T. gondii* parasites. Current was recorded for a maximum of 15 min per cell, digitized using an Axon Digidata 1550B (Molecular Devices, San Jose, CA) and the Axopatch software package (Molecular Devices, San Jose, CA). Conductance is calculated using Ohm's law, following correction for the pipette access resistance. Data analysis was performed offline using Clampfit 11.2 (Molecular Devices, San Jose, CA) and MATLAB R2022b (MathWorks, Natick, MA).

## Comparisons between optical and electrical transients

Raw fluorescence trajectories ($n = 22$; 160 s duration; $50 \times 50$ pixel ROI), with variable numbers of transients, were wavelet denoised (Daubechies db6 wavelet, empirical Bayes denoising threshold) and the time-varying background calculated on the denoised trajectory using the rolling ball algorithm in time (disc structural element, 2 s radius). This background was then subtracted from the raw fluorescence trajectory. In the absence of a transient, the background-subtracted baseline was characterized by an approximately zero mean Gaussian (0.56), indicating less than 1 intensity unit bias and consistent with appropriate background subtraction. To rapidly identify robust transients, a threshold of 5*(standard deviation) was used with MATLAB peakfinder; 30 transients were identified in this dataset, peak normalized, and the peak arbitrarily set to zero time (Fig. 3). Conductance transients were displayed similarly. Area under the curve (AUC) was calculated by summing the background-subtracted optical and conductance values from the start to the peak of the transient. The distributions around the mean normalized AUC for calcium and conductance transients were compared using a two-sample Kolmogorov–Smirnov test. Results reported in the text as not significant refer to comparisons in which there is insufficient evidence to reject the null hypothesis that the data came from populations with the same distribution.

## Data availability

This study includes no data deposited in external repositories.

The source data of this paper are collected in the following database record: biostudies:S-SCDT-10_1038-S44319-025-00564-9.

## Peer review information

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

## Acknowledgements

We thank Dr. Vern Carruthers for many helpful discussions on this project and members of the Ward lab for comments on the manuscript. This work was

supported by US Public Health Service grants U01AI169067 (GEW) and R01AI144369 (SL), European Research Council Advanced Grant 833309 (ML), and the intramural program of the NICHD. YK was supported by a JSPS research fellowship for Japanese Biomedical and Behavioral Researchers at NIH from 2021 to 2023.

## Author contributions

**Frances Male**: Conceptualization; Data curation; Software; Formal analysis; Investigation; Visualization; Methodology; Writing—original draft; Writing—review and editing. **Yuto Kegawa**: Conceptualization; Data curation; Formal analysis; Investigation; Visualization; Methodology; Writing—review and editing. **Paul S Blank**: Conceptualization; Data curation; Formal analysis; Supervision; Investigation; Visualization; Methodology; Writing—review and editing. **Irene Jiménez-Munguía**: Investigation; Writing—review and editing. **Saima M Sidik**: Resources; Writing—review and editing. **Dylan Valleau**: Resources; Writing—review and editing. **Sebastian Lourido**: Conceptualization; Resources; Supervision; Writing—review and editing. **Maryse Lebrun**: Conceptualization; Resources; Supervision; Visualization; Methodology; Writing—review and editing. **Joshua Zimmerberg**: Conceptualization; Resources; Data curation; Software; Formal analysis; Supervision; Funding acquisition; Visualization; Methodology; Project administration; Writing—review and editing. **Gary E Ward**: Conceptualization; Resources; Data curation; Software; Formal analysis; Supervision; Funding acquisition; Visualization; Methodology; Writing—original draft; Project administration; Writing—review and editing.

Source data underlying figure panels in this paper may have individual authorship assigned. Where available, figure panel/source data authorship is listed in the following database record: biostudies:S-SCDT-10_1038-S44319-025-00564-9.

## Disclosure and competing interests statement

The authors declare no competing interests.

# Expanded View Figures

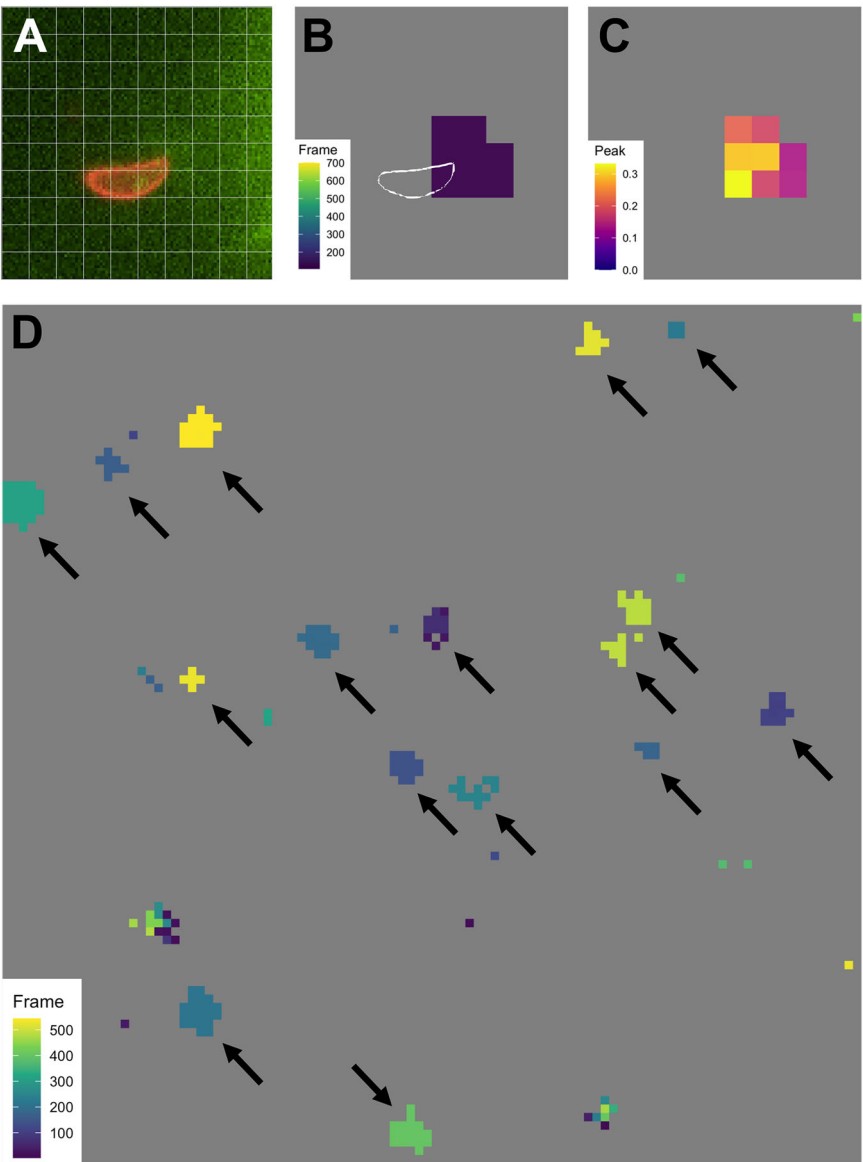

**Figure EV1.  Semi-automated workflow for quantifying the parasite-induced calcium transients in the host cell.**

(**A**) Individual invasion event used to illustrate the quantification scheme. Mean fluorescence intensity (MFI) was captured over time from 100 10 × 10 pixel regions of interest (ROIs; white grid lines) in this 100 × 100 pixel example field of view (FOV). The full FOV for each capture consists of 1020 × 1020 pixels. (**B**) PeakCaller output plotted back to ROI location and color-coded by frame (inset) at which the transient event reached maximal intensity; location of parasite outlined in white. This transient reached its peak at frame 106 of 700 frames total. (**C**) PeakCaller output plotted back to ROI location and color-coded by peak intensity (inset) achieved during the transient event. (**D**) Sixteen calcium transients (black arrows) were detected in this representative full FOV (221 × 221 µm) over the course of 48 s of recording (600 frames). Full captures consist of 1200 frames; fewer frames are shown here for clarity. MFI was captured over time from 10,404 10 × 10 pixel ROIs across 1020 × 1020 pixel FOV. PeakCaller output was plotted back to ROI location and color-coded by the frame (inset) when each transient reached peak intensity.

## A  WT (RH)

| Treatment | +Transient +Invasion | +Transient −Invasion | −Transient +Invasion | −Transient −Invasion |
|---|---|---|---|---|
| N/A | 206 | 3 | 9 | 836 |

## B  CLAMP

| Treatment | +Transient +Invasion | +Transient −Invasion | −Transient +Invasion | −Transient −Invasion | Fisher's exact test |
|---|---|---|---|---|---|
| DMSO | 521 | 9 | 57 | 1934 | $p < 0.0001$ |
| Rapamycin | 113 | 1 | 11 | 2412 | |

**B'**

| +Transient | −Transient | Fisher's exact test |
|---|---|---|
| 530 | 1991 | $p < 0.0001$ |
| 114 | 2423 | |

## C  FER2

| Treatment | +Transient +Invasion | +Transient −Invasion | −Transient +Invasion | −Transient −Invasion | Fisher's exact test |
|---|---|---|---|---|---|
| EtOH | 436 | 3 | 21 | 1165 | $p < 0.0001$ |
| ATc | 11 | 4 | 1 | 1852 | |

**C'**

| +Transient | −Transient | Fisher's exact test |
|---|---|---|
| 439 | 1186 | $p < 0.0001$ |
| 15 | 1853 | |

## D  Nd9

| Treatment | +Transient +Invasion | +Transient −Invasion | −Transient +Invasion | −Transient −Invasion | Fisher's exact test |
|---|---|---|---|---|---|
| EtOH | 261 | 1 | 56 | 711 | $p < 0.0001$ |
| ATc | 10 | 16 | 7 | 654 | |

**D'**

| +Transient | −Transient | Fisher's exact test |
|---|---|---|
| 262 | 767 | $p < 0.0001$ |
| 26 | 661 | |

## E  NdP1

| Treatment | +Transient +Invasion | +Transient −Invasion | −Transient +Invasion | −Transient −Invasion | Fisher's exact test |
|---|---|---|---|---|---|
| EtOH | 260 | 1 | 14 | 708 | $p < 0.0001$ |
| ATc | 42 | 29 | 6 | 1091 | |

**E'**

| +Transient | −Transient | Fisher's exact test |
|---|---|---|
| 261 | 722 | $p < 0.0001$ |
| 71 | 1097 | |

## F  RASP2

| Treatment | +Transient +Invasion | +Transient −Invasion | −Transient +Invasion | −Transient −Invasion | Fisher's exact test |
|---|---|---|---|---|---|
| EtOH | 384 | 2 | 23 | 1336 | $p < 0.0001$ |
| ATc | 18 | 6 | 0 | 902 | |

**F'**

| +Transient | −Transient | Fisher's exact test |
|---|---|---|
| 386 | 1359 | $p < 0.0001$ |
| 24 | 902 | |

**Figure EV2.   Total invasion and transient counts for each parasite line analyzed and contingency table analysis for the inducible knockdown parasites.**

(**A**) Total WT parasite counts were separated into four categories: +transient/+invasion; +transient/−invasion; −transient/+invasion; and −transient/−invasion. Each number represents the sum of three biological replicates, consisting of two to three technical replicates each. (**B**–**F**) Data from all mutant parasite lines categorized as in (**A**) within 4 × 2 contingency tables to compare control (top row) and protein-depleted (bottom row) parasites. Each number represents the sum of three biological replicates, consisting of two to three technical replicates each. Fisher's exact test was used for the comparison (right-most column). (**B'**–**F'**) Data from each mutant line (**B**–**F**) were summed based on the presence or absence of detected calcium transients (+transient, −transient) and organized into 2 × 2 contingency tables. Fisher's exact test was used for the comparison (right-most column).

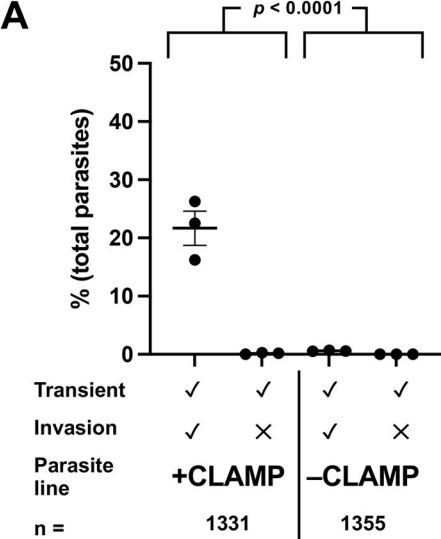

**A**

| | Transient | Invasion | Parasite line | n = |
|---|---|---|---|---|
| +CLAMP | ✓ / ✓ | ✓ / ✗ | +CLAMP | 1331 |
| −CLAMP | ✓ / ✓ | ✓ / ✗ | −CLAMP | 1355 |

*p* < 0.0001

**B**  CLAMP

| Treatment | +Transient +Invasion | +Transient −Invasion | −Transient +Invasion | −Transient −Invasion | Fisher's exact test |
|---|---|---|---|---|---|
| DMSO | 284 | 2 | 1 | 1044 | *p* < 0.0001 |
| Rapamycin | 8 | 0 | 0 | 1347 | |

**B'**

| Treatment | +Transient | −Transient | Fisher's exact test |
|---|---|---|---|
| DMSO | 286 | 1045 | *p* < 0.0001 |
| Rapamycin | 8 | 1347 | |

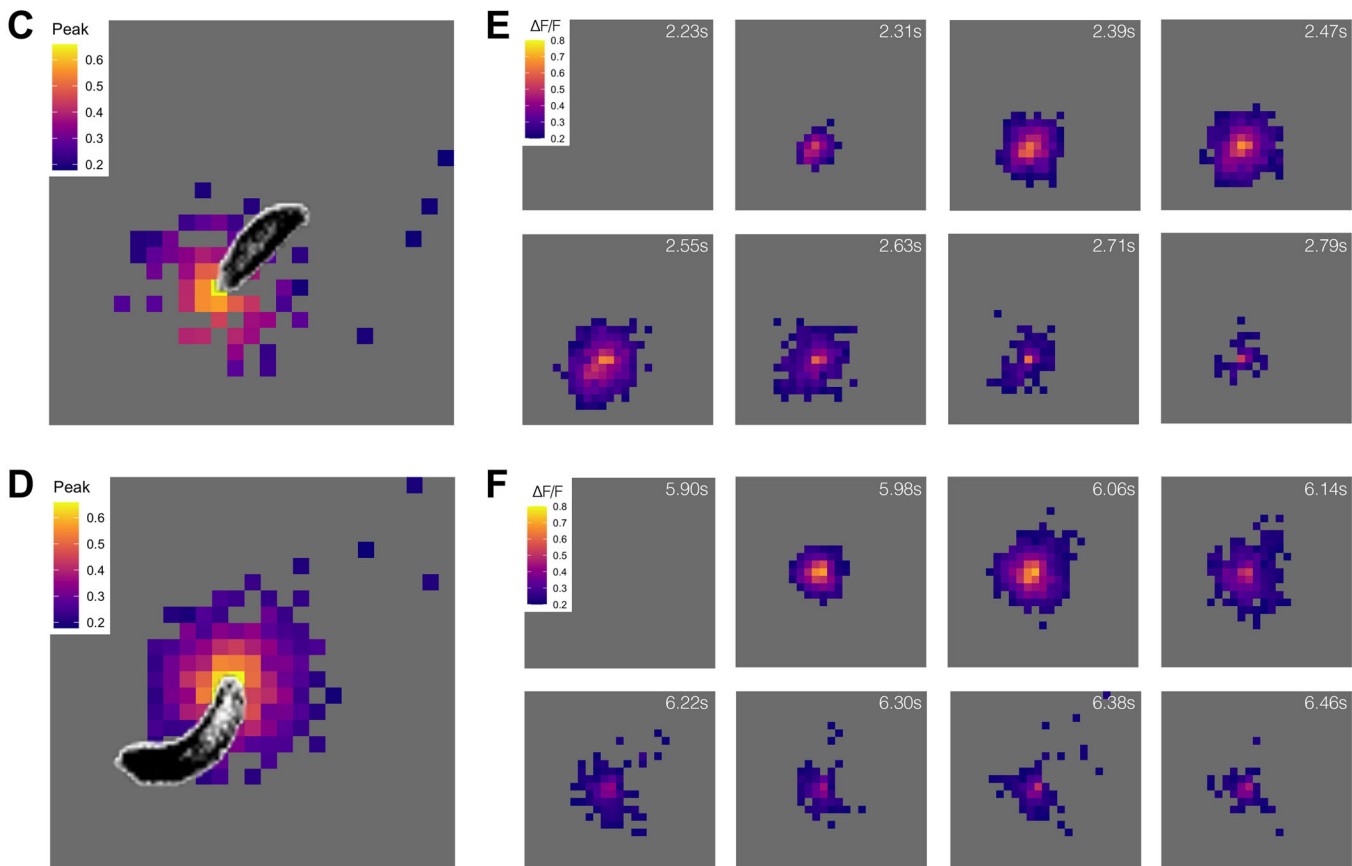

**Figure EV3. Improved detection of calcium transients.**

(A) Quantification of invasion events and calcium transients induced by control parasites (+CLAMP, $n = 1331$) compared to parasites depleted of CLAMP (−CLAMP, $n = 1355$) using a fluorescence detection method with increased sensitivity (Cal-520, probenecid treatment, 8 mM extracellular $CaCl_2$; see text for details). Each data point represents one biological replicate, consisting of the average of two to three technical replicates; horizontal bars indicate mean ± SEM. Comparison of total calcium transients between +CLAMP and −CLAMP groups was analyzed using Fisher's exact test, $p < 0.0001$ (Fig. EV3B'). (B) Total data from control-treated (top) parasites and CLAMP-depleted (bottom) parasites were categorized as in Fig. EV2A–F within a 4 × 2 contingency table for comparison. Fisher's exact test was used for the comparison (right-most column). Each number represents the sum of three biological replicates, consisting three technical replicates each. (B') Data from (B) were summed based on the presence or absence of detected calcium transients (+transient, −transient) and organized into 2 × 2 contingency tables. Fisher's exact test was used for the comparison (right-most column). (C–F) Panels (C, E) display the calcium transient shown in Movie EV4, after binning the data into 4 × 4 pixel ROIs (0.87 × 0.87 μm). Panel (C) shows the PeakCaller output based on mean fluorescence intensity within each ROI, color coded by peak intensity (see Fig. EV1 for PeakCaller methods). The overlay shows that location of the parasite at the initiation of invasion. Panel E shows the same transient, but in this case the change in fluorescence intensity within each ROI was divided by the median intensity within the ROI over the entire time series (ΔF/F). Time points represent consecutive frames and time is shown in seconds (s). Panels (D, F) display analysis of a second calcium transient (Movie EV5), analyzed identically to the data in panels (C, E).

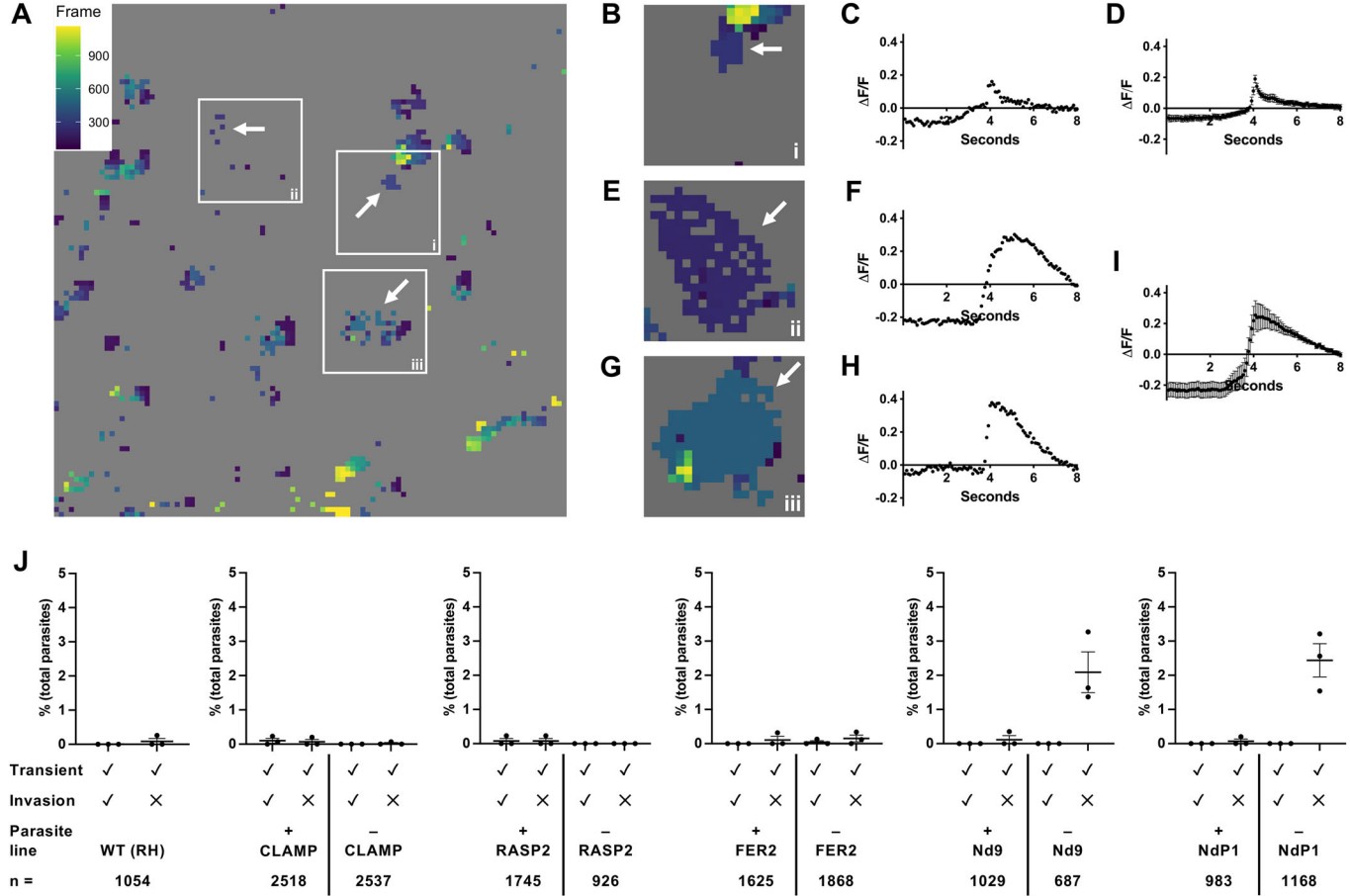

**Figure EV4. Aberrant calcium transients detected in all parasite lines tested are typically associated with non-invading parasites.**

(A) Calcium transients generated by NdP1-depleted parasites. The PeakCaller output from the full 1020 × 1020 pixel FOV was plotted back to ROI location and color-coded by frame (inset) at which the transients reached their maximal intensity. Box i highlights a calcium transient associated with an invading parasite and boxes ii and iii highlight two aberrant calcium transients associated with parasites that subsequently failed to invade. (B, E, G) Magnified views of the fluorescence signals detected in boxes i–iii from (A), after reanalyzing the data with parameters optimized for capture of aberrant transient events. (C, F, H) Quantification of Fluo-4 fluorescence levels (ΔF/F) in the host cell during the calcium transient events shown in (B, E, G) respectively. (D, I) Consensus plot of calcium transients generated by NdP1-depleted parasites that subsequently invaded the host cell ((D), $n = 15$) and NdP1-depleted parasites that did not subsequently invade ((I), $n = 9$). The fluorescence intensities in the 100 frames surrounding the peak of each calcium transient were averaged across all transients, the peaks of which were aligned to frame 51. The plot shows the mean ± SEM at each time point. (J) Quantification of the frequency of aberrant spike generation by each of the parasite lines analyzed in this study. Each graph shows the number of aberrant spikes generated, as a percentage of the total number of parasites counted, and whether the aberrant spikes were associated with invading or non-invading parasites. Each data point represents one biological replicate, consisting of the average of two to three technical replicates. Horizontal bars indicate mean ± SEM.

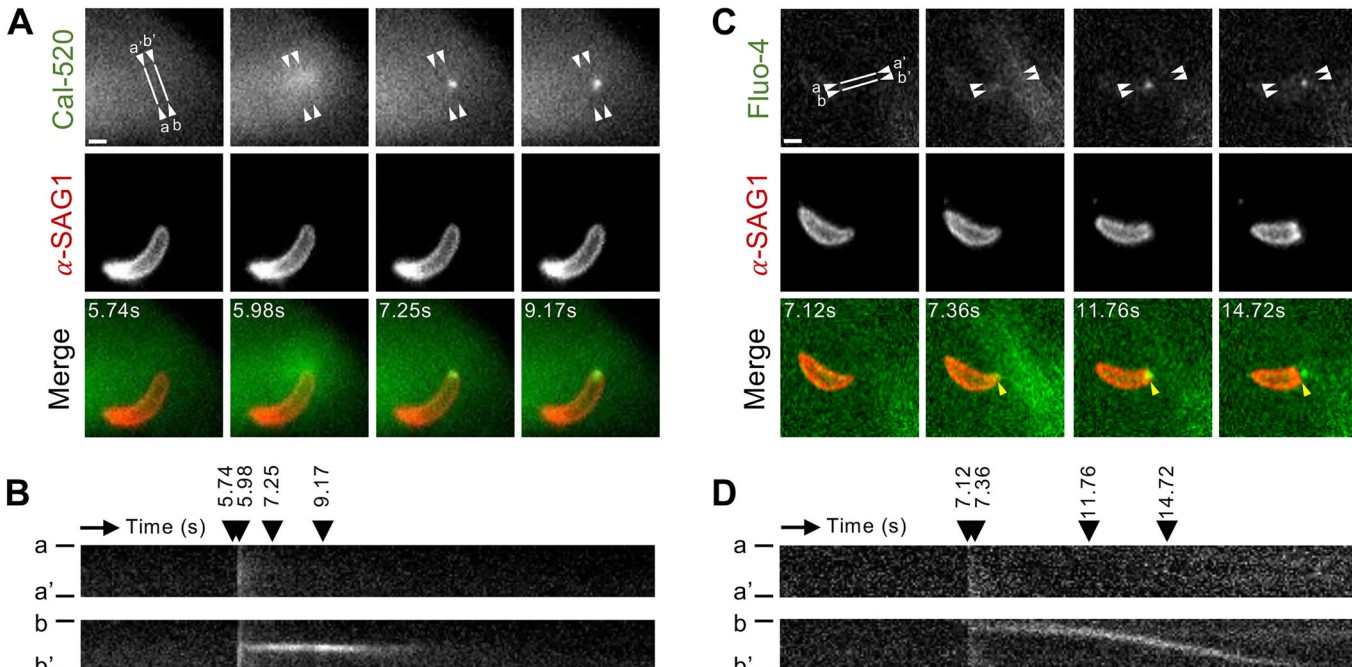

**Figure EV5. Development of a dot of fluorescence at the apical end of parasites invading calcium indicator-loaded host cells.**

Cal-520 (**A**) or Fluo-4 (**C**) fluorescence (top row) during invasion of calcium indicator-loaded host cells by anti-SAG1-labeled parasites (middle row). Note the appearance of a dot of fluorescence at the parasite apex. Scale bar = 2 μm; time is shown in seconds (s). Yellow arrowheads in (**C**) indicate the position of the moving junction. The full videos from which these images were extracted are presented as Movies EV5 and EV3, respectively. Panels (**B, D**) show the kymographs of calcium indicator fluorescence intensity, captured over time along the lines (a to a' and b to b') indicated in Panels (**A, C**). For both, a to a' is a line through a control region of the cell, while b to b' captures the appearance/movement of the dot. In kymograph (**B**) the calcium transient occurs at 5.98 s and precedes the formation and intensification of the dot of fluorescence that is evident along line b to b'. In kymograph (**D**), the calcium transient is evident at 7.36 s, and the subsequent movement of the fluorescent dot at the apical tip of the parasite past the moving junction (yellow arrowhead) and into the host cell is evident along line b to b'.

