## [Peer Review File · EMBO Reports]

Perforation of the host cell plasma membrane during *Toxoplasma* invasion requires rhoptry exocytosis

Frances Male, Yuto Kegawa, Paul Blank, Irene Jimenez Munguia, Saima Sidik, Dylan Valleau, Sebastian Lourido, Maryse Lebrun, Joshua Zimmerberg, and Gary Ward

Corresponding authors: Gary Ward (Gary.Ward@uvm.edu) , Joshua Zimmerberg (zimmerbj@mail.nih.gov)

Review Timeline:

Submission Date:	27th Nov 24
Editorial Decision:	28th Jan 25
Revision Received:	12th May 25
Editorial Decision:	30th Jun 25
Revision Received:	10th Jul 25
Accepted:	1st Aug 25

Transaction Report:

Dear Gary,

Thank you for the submission of your research manuscript to our journal. I apologize again for the delay in handling your manuscript. As you know, we have received meanwhile two referee reports on your manuscript and I have decided to proceed with these (copied below).

As you will see, both referees acknowledge that the findings are interesting and that the conclusions are overall supported by the data presented but they also raise a number of concerns. I think the most critical concern relates to the kinetics of Ca²⁺ pulses compared to the extremely fast changes in membrane conductance observed by patch clamp. Referee 2 is concerned that these two methods detect two different, potentially unrelated events and that calcium pulses do not necessarily represent early perforation events but rather a later invasion event. While we do not require mechanistic insights for publication at EMBO Reports (stated but not mandated by referee 1), this concern must be addressed in a revision.

The contribution of mechanical forces should also be discussed as well as other means that might contribute to ROP protein translocation in addition to perforation (referee 1). All other technical concerns need to be addressed.

Given the constructive comments, we would like to give you the chance to revise your manuscript with the understanding that the referee concerns (as detailed above and in their reports) must be fully addressed and their suggestions taken on board. Please address all referee concerns in a complete point-by-point response. Acceptance of the manuscript will depend on a positive outcome of a second round of review. It is EMBO Reports policy to allow a single round of revision only and acceptance or rejection of the manuscript will therefore depend on the completeness of your responses included in the next, final version of the manuscript.

We realize that it is difficult to revise to a specific deadline. In the interest of protecting the conceptual advance provided by the work, we recommend a revision within 3 months (April 28). Please discuss the revision progress ahead of this time with the editor if you require more time to complete the revisions.

I am also happy to discuss the revision further via e-mail or a video call, if you wish.

*****IMPORTANT NOTE:

We perform an initial quality control of all revised manuscripts before re-review. Your manuscript will FAIL this control and the handling will be delayed IN CASE the following APPLIES:

- 1) A data availability section providing access to data deposited in public databases is missing. If you have not deposited any data, please add a sentence to the data availability section that explains that.
- 2) Your manuscript contains statistics and error bars based on n=2. Please use scatter blots in these cases. No statistics should be calculated if n=2.

When submitting your revised manuscript, please carefully review the instructions that follow below. Failure to include requested items will delay the evaluation of your revision.*****

- 1) a .docx formatted version of the manuscript text (including legends for main figures, EV figures and tables). Please make sure that the changes are highlighted to be clearly visible.
- 2) individual production quality figure files as .eps, .tif, .jpg (one file per figure). Please download our Figure Preparation Guidelines (figure preparation pdf) from our Author Guidelines pages <https://www.embopress.org/page/journal/14693178/authorguide> for more info on how to prepare your figures.
- 3) a .docx formatted letter INCLUDING the reviewers' reports and your detailed point-by-point responses to their comments. As part of the EMBO Press transparent editorial process, the point-by-point response is part of the Review Process File (RPF), which will be published alongside your paper.
- 4) a complete author checklist, which you can download from our author guidelines (<<https://www.embopress.org/page/journal/14693178/authorguide>>). Please insert information in the checklist that is also reflected in the manuscript. The completed author checklist will also be part of the RPF.
- 5) Please note that all corresponding authors are required to supply an ORCID ID for their name upon submission of a revised

manuscript (<<https://orcid.org/>>). Please find instructions on how to link your ORCID ID to your account in our manuscript tracking system in our Author guidelines (<<https://www.embopress.org/page/journal/14693178/authorguide#authorshipguidelines>>)

6) We replaced Supplementary Information with Expanded View (EV) Figures and Tables that are collapsible/expandable online. A maximum of 5 EV Figures can be typeset. EV Figures should be cited as 'Figure EV1, Figure EV2' etc... in the text and their respective legends should be included in the main text after the legends of regular figures.

7) Please note that a Data Availability section at the end of Materials and Methods is now mandatory. In case you have no data that requires deposition in a public database, please state so instead of refereeing to the database. See also < <https://www.embopress.org/page/journal/14693178/authorguide#dataavailability>>. Please note that the Data Availability Section is restricted to new primary data that are part of this study.

Additional information on source data and instruction on how to label the files are available <<https://www.embopress.org/page/journal/14693178/authorguide#sourcedata>>.

10) Figure legends and data quantification:
The following points must be specified in each figure legend:

- the name of the statistical test used to generate error bars and P values,
 - the number (n) of independent experiments (please specify technical or biological replicates) underlying each data point,
 - the nature of the bars and error bars (s.d., s.e.m.)
- If the data are obtained from n {less than or equal to} 5, show the individual data points in addition to the SD or SEM.
- If the data are obtained from n {less than or equal to} 2, use scatter blots showing the individual data points.

See also the guidelines for figure legend preparation:
<https://www.embopress.org/page/journal/14693178/authorguide#figureformat>

11) Our journal encourages inclusion of *data citations in the reference list* to directly cite datasets that were re-used and obtained from public databases. Data citations in the article text are distinct from normal bibliographical citations and should directly link to the database records from which the data can be accessed. In the main text, data citations are formatted as follows: "Data ref: Smith et al, 2001" or "Data ref: NCBI Sequence Read Archive PRJNA342805, 2017". In the Reference list, data citations must be labeled with "[DATASET]". A data reference must provide the database name, accession number/identifiers and a resolvable link to the landing page from which the data can be accessed at the end of the reference. Further instructions are available at <<https://www.embopress.org/page/journal/14693178/authorguide#referencesformat>>.

12) All Materials and Methods need to be described in the main text using our 'Structured Methods' format. According to this format, the Methods section includes a Reagents and Tools Table (listing key reagents, experimental models, software and relevant equipment and including their sources and relevant identifiers) followed by a Methods and Protocols section describing the methods, ideally using a step-by-step protocol format. The aim is to facilitate adoption of the methodologies across labs. Please download and fill our Reagents and Tools Table template (.docx), which you can find in our author guidelines:

13) As part of the EMBO publication's Transparent Editorial Process, EMBO Reports publishes online a Review Process File to accompany accepted manuscripts. This File will be published in conjunction with your paper and will include the referee reports, your point-by-point response and all pertinent correspondence relating to the manuscript.

Kind regards,

Martina

=====

Referee #1:

This study addresses an important and long-standing question in apicomplexan biology that is also of interest for microbiologists: how do proteins secreted by a pathogen cross the host cell membrane when this would normally be a barrier for translocation? Bacteria and viruses have solved this problem in several ways, but the authors point that such systems are not conserved in parasites, hence they must use a different mechanism. They approach the problem by developing assays to monitor the transient disruption of the host membrane during host cell invasion by *Toxoplasma*. To do so, they utilize previously described mutants in *rhoptyr* and apical vesicle protein fusion that block protein translocation and cell invasion. The strength of the study is the time resolved, spatial detection of membrane permeation that requires these previously characterized proteins and the studies are convincing and highly insightful. There are a number of modifications the authors should consider making to improve their manuscript prior to publication.

Main findings, potential impact, and weaknesses

The primary observation made by the study is that a transient permeabilization of the host cell plasma membrane happens prior to invasion and this event requires proteins that mediate *rhoptyr* AV fusion and AV-plasma membrane fusion. These data are thoroughly convincing and provide an important advance by connecting previously known mutant phenotypes (eg secretion and invasion) to physiological events that occur prior to invasion. Not shown is the mechanism for how host membrane permeabilization occurs, nor is it conclusively shown that this is the primary means for ROP protein translocation. If it is, then translocation would need to be extremely fast and occur in single burst. The kinetics of this process have not been examined, nor can it be ruled out that some proteins might be translocated after the transient permeabilization heals, which it evidently does quite rapidly. Overall I found the manuscript interesting and appreciated that the findings were not overstated; instead the authors are careful to pose some of their main conclusions as hypotheses.

1) In the Introduction, examples are given of bacterial machinery for protein translocation and the point is made that these are not conserved in parasites. However, colicins and viral fusion proteins also cross membranes by more direct routes. Could similar mechanisms contribute to the transfer of secreted ROPs or are the authors convinced they do not have appropriate properties i.e they are not cationic peptides or arginine rich transfer proteins?

2) Lines 46-48 in the Introduction seem to imply that the only proteins translocated into the host cell are derived from *rhoptyries*. In fact, some dense granule proteins also are translocated into the host cell, albeit by a completely different mechanism. Without going into excessive detail, this other pathway should be mentioned.

- 3) The patch clamp example in Figure 1 appears to be a single reading. It would be appropriate to include some analysis of the range and average values for amplitude and duration for multiple events. The duration is of particular interest in relation to the hypothesis that ROP proteins must enter the cell during this permeated state. Alternatively, they could include a summary of the findings in the companion paper as this would provide better context for the proposed role of membrane disruption.
- 4) The authors argue that AVs or rhoptries likely contain some agent that breaches the membrane, presumably in a detergent like manner. Alternatively, can they rule out that the breach is due to a mechanical force due to extrusion of the conoid? Do all the mutants studied here extend the conoid to the same extent, for example under calcium ionophore stimulation? This would be easy to test and in the absence of such data, the alternative should be acknowledged, even if it seems less likely.
- 5) Several groups have shown that ROP proteins can be directly secreted into host cells where they have fates independent of invading parasites (e.g. Hakansson 2001 and Koshy 2012). These studies are worth citing as they demonstrate that ROP secretion across the PM can occur without invasion, thus uncoupling the processes that the authors argue are linked in the present study. It would be interesting to determine if the transient membrane breach also occurs in these examples. Did the authors observe transient disruptions that were not accompanied by invasion? If not, please speculate on why they were missed.
- 6) The citation of Figure 1A in the Introduction is somewhat unconventional. An alternative would be to greatly reduce this final paragraph and simply set the stage for the question being tested, rather than summarizing the findings. The mutants could be introduced at a later stage when they are actually employed.

Referee #2:

The paper by Male et al explores the nature of the perforation events required for the invasion of *T. gondii* to host cells. The authors analyze the early events upon contact of tachyzoites with the host cell observing an increase in membrane conductance (described in detail in the manuscript by Kegawa et al) as well as a later increase in calcium that correlates with the invasion. While the use of rhoptry secretion mutants provide support to the idea that this is required for permeation, there are some technical issues that question the interpretation of the data:

- Due to the time-consuming and low throughput nature of the electrophysiological recordings, the authors propose to use Fluo-4 calcium imaging to monitor the invasion events and use it as a proxy for permeation. But the temporal resolution of the calcium measurements in Fig. 2 is in the order of seconds while the changes in membrane conductance detected by patch clamp are extremely fast (ms) and suggest a quick re-sealing of the membrane. So, the two methods are most probably measuring different events: one is the fast conductance transient and another in the invasion itself. How do you reconcile the difference in timing of these events? They could be unrelated, so you are not really monitoring the perforation event. In supplementary video 1, the wave of calcium in the host cell seems to precede the highly localized calcium signal that appears at the apical end of the parasite. This also suggests two different events. This is problematic, because all the further experiments are assuming that the calcium transients are representing the early permeation events but that is not necessarily true. The results in the companion paper by Male et al also argue against calcium being the main ion responsible for the conductance transients. Some further analysis and clarification is needed to understand and maybe de-couple both events.
- For Figure 2 F, explain calcium-free conditions.
- Figure 3 is suggestive of two correlated events, but is no demonstration that the calcium influx is happening through the same perforation pore (also argued against by the results of Male et al)
- Do CLAMP and other mutants affect the transients detected by patch-clamp? This is necessary to demonstrate that they are affecting perforation and that the reduction in calcium transients is not derived from the lack of invasion.
- You are still not able to separate the conductance transient from the perforation event and the invasion. Could this be analyzed with a mutant that gets stuck, and is able to perforate but not invade?

Dear Dr. Rembold:

Thank you for forwarding the reviewer critiques of our manuscript. We were grateful for the reviewers' positive comments, and we have addressed their concerns (underlined below) in the revised manuscript as follows:

Referee #1:

This study addresses an important and long-standing question in apicomplexan biology that is also of interest for microbiologists: how do proteins secreted by a pathogen cross the host cell membrane when this would normally be a barrier for translocation? Bacteria and viruses have solved this problem in several ways, but the authors point that such systems are not conserved in parasites, hence they must use a different mechanism. They approach the problem by developing assays to monitor the transient disruption of the host membrane during host cell invasion by *Toxoplasma*. To do so, they utilize previously described mutants in rhoptry and apical vesicle protein fusion that block protein translocation and cell invasion. The strength of the study is the time resolved, spatial detection of membrane permeation that requires these previously characterized proteins and the studies are convincing and highly insightful. There are a number of modifications the authors should consider making to improve their manuscript prior to publication.

Main findings, potential impact, and weaknesses

The primary observation made by the study is that a transient permeabilization of the host cell plasma membrane happens prior to invasion and this event requires proteins that mediate rhoptry AV fusion and AV-plasma membrane fusion. These data are thoroughly convincing and provide an important advance by connecting previously known mutant phenotypes (eg secretion and invasion) to physiological events that occur prior to invasion. Not shown is the mechanism for how host membrane permeabilization occurs, nor is it conclusively shown that this is the primary means for ROP protein translocation. If it is, then translocation would need to be extremely fast and occur in single burst. The kinetics of this process have not been examined, nor can it be ruled out that some proteins might be translocated after the transient permeabilization heals, which it evidently does quite rapidly. Overall I found the manuscript interesting and appreciated that the findings were not overstated; instead the authors are careful to pose some of their main conclusions as hypotheses.

1) In the Introduction, examples are given of bacterial machinery for protein translocation and the point is made that these are not conserved in parasites. However, colicins and viral fusion proteins also cross membranes by more direct routes. Could similar mechanisms contribute to the transfer of secreted ROPs or are the authors convinced they do not have appropriate properties i.e they are not cationic peptides or arginine rich transfer proteins?

> Because characterization of the perforation is the focus of the companion paper (Kegawa, Male et al), we and our co-authors have discussed the potential role of known pore-forming proteins there rather than here. As to the specific examples raised by the reviewer, cell-penetrating peptides would have to be present at very high concentrations in the rhoptries to enable cargo translocation, and there is no evidence from rhoptry proteomics studies that this is the case (Peter Bradley, personal communication). Colicin causes cell death through irreversible perforation, so if such a protein is involved there will also need to be a regulatory component that removes or inactivates the pore after translocation of the rhoptry effector proteins. Identifying the perforating agent and any accessory proteins involved in its opening, closing or regulation is clearly the next priority for us, as stated on lines 627-628 and 698-700*.

* line numbers refer to the version of the manuscript with changes tracked.

2) Lines 46-48 in the Introduction seem to imply that the only proteins translocated into the host cell are derived from rhoptries. In fact, some dense granule proteins also are translocated into the host cell, albeit by a completely different mechanism. Without going into excessive detail, this other pathway should be mentioned.

> We agree; see lines 64-75 and 99-103 in the revised manuscript

3) The patch clamp example in Figure 1 appears to be a single reading. It would be appropriate to include some analysis of the range and average values for amplitude and duration for multiple events. The duration is of particular interest in relation to the hypothesis that ROP proteins must enter the cell during this permeated state. Alternatively, they could include a summary of the findings in the companion paper as this would provide better context for the proposed role of membrane disruption.

> We agree; see lines 170-173. We have also added additional text (lines 559-560, 685-691) to better explain that while the decrease in conductance and calcium influx after the peak may reflect gated closure of the pore, it could alternatively correspond to occlusion of the pore by translocating cargo such as rhoptry proteins, reducing the influx of calcium and relevant charge-carrying ions. In this latter scenario, effector translocation (and persistence of the perforation) would continue beyond the short duration of the conductance/calcium transients we detect in our assays.

4) The authors argue that AVs or rhoptries likely contain some agent that breaches the membrane, presumably in a detergent like manner. Alternatively, can they rule out that the breach is due to a mechanical force due to extrusion of the conoid? Do all the mutants studied here extend the conoid to the same extent, for example under calcium ionophore stimulation? This would be easy to test and in the absence of such data, the alternative should be acknowledged, even if it seems less likely.

> The suggested experiment has been done previously by others for at least two of the mutants: both Nd9- and FER2-deficient parasites lack the ability to generate spikes

(Figure 5 in our manuscript) but nevertheless support calcium ionophore-induced conoid extension (Aquilini et al 2021, Coleman et al 2018). It is also difficult to imagine how mechanical force exerted by the conoid on the host cell membrane would produce the quantal steps in conductance reported in our companion paper (Kegawa, Male et al). However, we cannot rule out membrane tension induced by conoid extension plays some role in either inserting the pores or regulating their function, and we have added a brief discussion of this possibility on lines 629-641.

5) Several groups have shown that ROP proteins can be directly secreted into host cells where they have fates independent of invading parasites (e.g. Hakansson 2001 and Koshy 2012). These studies are worth citing as they demonstrate that ROP secretion across the PM can occur without invasion, thus uncoupling the processes that the authors argue are linked in the present study. It would be interesting to determine if the transient membrane breach also occurs in these examples. Did the authors observe transient disruptions that were not accompanied by invasion? If not, please speculate on why they were missed.

> Figures 5 and 6 show that the calcium transients can indeed occur without subsequent invasion, and the specific frequencies with which this occurs in the different parasite lines tested are reported in Figures EV2 and EV3. Furthermore, RON2-deficient parasites have been shown to translocate rhoptry effector proteins without invasion (Lamarque et al 2014), and we show in the companion paper that these parasites generate both conductance and calcium transients at the same frequency as wild-type parasites (Kegawa, Male et al).

6) The citation of Figure 1A in the Introduction is somewhat unconventional. An alternative would be to greatly reduce this final paragraph and simply set the stage for the question being tested, rather than summarizing the findings. The mutants could be introduced at a later stage when they are actually employed.

> We agree; we have moved the schematic to Figure 4A and deleted reference to it from the Introduction, as suggested.

Referee #2:

The paper by Male et al explores the nature of the perforation events required for the invasion of *T. gondii* to host cells. The authors analyze the early events upon contact of tachyzoites with the host cell observing an increase in membrane conductance (described in detail in the manuscript by Kegawa et al) as well as a later increase in calcium that correlates with the invasion. While the use of rhoptry secretion mutants provide support to the idea that this is required for permeation, there are some technical issues that question the interpretation of the data:

1) Due to the time-consuming and low throughput nature of the electrophysiological recordings, the authors propose to use Fluo-4 calcium imaging to monitor the invasion events and use it as a proxy for permeation. But the temporal resolution of the calcium measurements in Fig. 2 is in the order of seconds while the changes in membrane conductance detected by patch clamp are extremely fast (ms) and suggest a quick re-sealing of the membrane. So, the two methods are most probably measuring different events: one is the fast conductance transient and another in the invasion itself. How do you reconcile the difference in timing of these events? They could be unrelated, so you are not really monitoring the perforation event.

> Are the two assays detecting the same perforation event? We agree with the reviewer about the importance of this question, which we have considered very carefully.

First, to clarify: the duration of a process is not the same as the temporal resolution of the assay that is used to detect the process. The temporal resolution of the fluorescence experiment is tens of msec; slower than the electrophysiology (sub msec) but not seconds as the reviewer has suggested.

Second, given the enormous calcium electrochemical gradient across the host cell plasma membrane, we would expect calcium influx through any but the most highly selective pores. We note that even the Sec61 channel in the ER membrane, one of the most well studied and tightly regulated protein translocons, allows some calcium leakage into the cytosol down the steep ER to cytosol calcium gradient (Lang et al 2017). Furthermore, in the absence of extracellular calcium the calcium transients are not observed.

Third, we see exactly one conductance transient and one calcium transient per invasion event. This rules out the possibility that the conductance transient occurs first and triggers a subsequent influx of calcium, as the later calcium influx would have been detectable as a second conductance transient (calcium ions carrying the charge in this case). A second conductance transient during invasion was never observed.

Fourth, we now present data showing that the calcium transient is first detected in the host cell within a small (800x800nm) region immediately adjacent to the point of parasite attachment, and with time spreads radially outward from that point (new EV3 C-F) with kinetics consistent with a simple diffusion model. The calcium transient is therefore dependent on extracellular calcium, initiated in a highly localized manner, and spreads within the host cell in a manner consistent with simple diffusion. Together, these data are more consistent with a single perforation event than an initial perforation followed by secondary signal-mediated calcium release from internal stores.

Fifth, we would not expect the duration of the conductance and calcium transients to be the same, even if (as our data suggest) they are reporting on the same event. The conductance transient is a direct measurement of ion flow through the perforation(s) in the membrane; when the perforation is open ions flow and when the perforation is closed flow stops. The fluorescence-based assay is specifically measuring the number of calcium ions that flow through this same perforation. However, in contrast to patch clamping, which is directly measuring ion flow, the calcium indicator provides a chemical signal which, given the off and on rates of the indicator, diffusion of calcium and calcium-bound indicator from the site of calcium entry, and calcium sequestration mechanisms of the cell, would

almost certainly result in a longer duration signal than electrophysiological recordings of conductance, which is what we see.

Sixth, the two signals are not unrelated, as demonstrated by the indistinguishable distributions of the magnitudes of the calcium and conductance signals during the rising phase of the transient, when the diffusion and sequestration effects on the calcium signal are expected to be least pronounced (new Figure 3C). The similar variation in properties of the signals shows they are indeed correlated. Consistent with this observation, three different parasite lines (wild-type and parasites lacking RASP2 or RON2) and a fourth not reported in our companion paper (Ndp1-depleted) all gave concordant results: the RASP2- and Ndp1-depleted parasites generated fewer conductance and calcium transients than wildtype, the RON2-depleted parasites did not.

The most parsimonious explanation for all the data summarized above is that the two assays are reporting on the same event. We have extended our discussion of this important question in both the Results section and the Discussion (lines 242-252, lines 542-572).

2) In supplementary video 1, the wave of calcium in the host cell seems to precede the highly localized calcium signal that appears at the apical end of the parasite. This also suggests two different events.

> We believe these are indeed two different events. The most likely explanation for the appearance of the fluorescent dot at the apical end of the parasite is that the parasite perforates the host cell, calcium flows into the host cell, and concomitantly or shortly thereafter Fluo4 diffuses in the opposite direction through the perforation into an apical parasite compartment, perhaps the exocytosed rhoptries. The diffusing Fluo4 may have calcium bound to it or may encounter high calcium levels within the parasite, in either case generating brightly fluorescent dot at the parasite apex. So yes: we believe these are two different events separated in time, but both likely result from perforation of the host cell membrane. We have tried to explain this more clearly on lines 650-675, and we have now included kymograph data that demonstrate that the fluorescent dot develops after the initial calcium transient (Figure EV5 A) and moves into the host cell during invasion, presumably at the apical tip of the parasites (Figure EV5 B).

3) This is problematic, because all the further experiments are assuming that the calcium transients are representing the early permeation events but that is not necessarily true. The results in the companion paper by Male et al also argue against calcium being the main ion responsible for the conductance transients. Some further analysis and clarification is needed to understand and maybe de-couple both events.

> Neither paper argues that calcium is the charge-carrying ion in the conductance experiments; in fact the data in Kegawa, Male et al show otherwise as the reviewer notes (and as we have now stated even more clearly on lines 585-589). The data show that the perforation is not dependent on extracellular calcium; calcium influx is simply an experimentally accessible way to detect the presence of the perforation because of the

steep electrochemical gradient of calcium across the host plasma membrane and because fast and sensitive indicators are available to detect it. Other charge-carrying ions are also clearly flowing down their electrochemical gradients into or out of the cell; otherwise we would not have observed conductance transients of similar size and shape in medium depleted of external calcium, as we report in Kegawa, Male et al.

4) For Figure 2 F, explain calcium-free conditions.

> Done; see lines 790-794.

5) Figure 3 is suggestive of two correlated events, but is no demonstration that the calcium influx is happening through the same perforation pore (also argued against by the results of Male et al)

> See response above to the suggestion that the two events might be unrelated. Nothing in the companion paper argues against the idea that the two assays are detecting the same perforation event.

6) Do CLAMP and other mutants affect the transients detected by patch-clamp? This is necessary to demonstrate that they are affecting perforation and that the reduction in calcium transients is not derived from the lack of invasion.

> Testing all the mutants analyzed here in the patch clamp assay would have been an enormous task which is precisely why we developed the alternative, higher throughput calcium influx assay. That said, we did test selected mutants of particular interest (RON2, RASP2) in both assays, confirming that the two assays gave concordant results.

Note also that the RON2 mutants presented in the companion paper show clearly that non-invading parasites can generate conductance transients at ~WT levels.

7) You are still not able to separate the conductance transient from the perforation event and the invasion. Could this be analyzed with a mutant that gets stuck, and is able to perforate but not invade?

> The conductance transient IS the perforation event so we do not expect to be able to separate them. As described above, Figures 5 and 6 show that the calcium transients can indeed occur without subsequent invasion, and the specific frequencies with which this occurs in the different parasite lines tested are reported in Figures EV2 and EV3. This observation is not at all surprising -- and is consistent with the translocation hypothesis -- since we know from the work of Lamarque et al 2014 that exocytosis and effector translocation occur in RON2-deficient parasites, and we show in the companion paper that these parasites generate both conductance and calcium transients at the same frequency as wild-type parasites (Kegawa, Male et al).

Dear Dr. Ward

Thank you once more for the submission of your revised manuscript to EMBO reports. I have already forwarded the referee reports to you and you find them again copied below.

As you know, both referees support publication in EMBO reports after some minor revisions.

From the editorial side, there are also a few things that we need before we can proceed with the official acceptance of your study.

- Please provide up to 5 keywords.
- Please update the reference to Kegawa and Male et al to EMBO reports in press.
- Reference to preprints need to be labeled in the text as (preprint: Gui et al., 2022) and in the reference list with a [PREPRINT] at the end of the reference.
- References: et al needs to be used after 10 author names; DOIs should only be used for preprints and datasets that have not been published yet.
- Reagents and Tools table: Please remove the "Instructions" paragraph. You can also delete all sections that are not required, such as Recombinant DNA etc.
- Please provide the conflict of interest statement in a separate 'Disclosure and competing interests statement' paragraph. For more information see <https://www.embopress.org/page/journal/14693178/authorguide#conflictsofinterest>
- Appendix: please add page numbers to the table of content.
- Movies: when each movie is played, the title/name that is displayed is Supp_Video 1, etc. instead of Movie EV#1. Please correct this.
- Figure legends need to be placed at the end, after the References.
- Please remove the Appendix figure legends and the movie figure legends from the manuscript.
- Statistical test information in the figure legends: please specify the exact p values in the legends of figures 4C, 5A-C; 6A, EV3 A, B, B'.
- Please upload the source data as one zip folder per figure. Thank you.
- Finally, EMBO Reports papers are accompanied online by
 - A) a short (1-2 sentences) summary of the findings and their significance,
 - B) 2-3 bullet points highlighting key results and
 - C) a schematic summary figure that provides a sketch of the major findings (not a data image).Please provide the summary figure as a separate file in PNG or JPG format at a size of 550x300-600 pixels (width x height). Please note that the size is rather small and that text needs to be readable at the final size. Please send us this information along with the revised manuscript.

With kind regards,

Martina

=====

Referee #1:

The authors have done an adequate job of addressing my prior concerns with one exception listed below. The answer to this question seems to be about something entirely different (related to pore closure or block). I am asking about replicates-averages, raneg etc.,).

3) The patch clamp example in Figure 1 appears to be a single reading. It would be appropriate to include some analysis of the range and average values for amplitude and duration for multiple events. The duration is of particular interest in relation to the hypothesis that ROP proteins must enter the cell during this permeated state.

Referee #2:

Thank you for the careful discussion of the questions raised in the previous revision of the manuscript. This work in conjunction with the companion paper by Kegawa et al. moves us closer to understanding the events that regulate the invasion of cells by *T. gondii*.

The authors addressed the remaining editorial issues.

Gary Ward
University of Vermont Larner College of Medicine
Microbiology and Molecular Genetics
95 Carrigan Drive
316 Stafford Hall
Burlington, VT 05405
United States

Dear Gary,

Thank you very much for implementing the final minor revisions. Now all looks good and I am very pleased to accept your manuscript for publication in the next available issue of EMBO reports. Thank you for your contribution to our journal.

Best regards,

Martina
